# Thermodynamic signatures of the field-induced states of graphite

D. LeBoeuf[1], C.W. Rischau[2], G. Seyfarth[1,3], R. Küchler[4], M. Berben[5], S. Wiedmann[5], W. Tabis[1,6], M. Frachet [1], K. Behnia[2] & B. Fauqué[2,7]

When a magnetic field confines the carriers of a Fermi sea to their lowest Landau level, electron−electron interactions are expected to play a significant role in determining the electronic ground state. Graphite is known to host a sequence of magnetic field-induced states driven by such interactions. Three decades after their discovery, thermodynamic signatures of these instabilities are still elusive. Here we report the detection of these transitions with sound velocity measurements. The evolution of elastic constant anomalies with temperature and magnetic field allows to draw a detailed phase diagram which shows that the ground state evolves in a sequence of thermodynamic phase transitions. Our analysis indicates that the electron−electron interaction is not the sole driving force of these transitions and that lattice degrees of freedom play an important role.

[1] Laboratoire National des Champs Magnétiques Intenses (LNCMI-EMFL), CNRS, UGA, UPS, INSA, Grenoble/Toulouse, France. [2] ESPCI ParisTech, PSL Research University; CNRS; Sorbonne Universités, UPMC Univ. Paris 6, LPEM, 10 rue Vauquelin, F-75231 Paris Cedex 5, France. [3] Université Grenoble-Alpes, Grenoble, France. [4] Max Planck Institute for Chemical Physics of Solids, Nöthnitzer Str. 40, 01187 Dresden, Germany. [5] High Field Magnet Laboratory (HFML-EMFL) and Institute for Molecules and Materials, Radboud University, Toernooiveld 7, 6525 ED Nijmegen, The Netherlands. [6] AGH University of Science and Technology, Faculty of Physics and Applied Computer Science, 30-059 Krakow, Poland. [7] JEIP, USR 3573 CNRS, Collège de France, PSL Research University, 11, place Marcelin Berthelot, 75231 Paris Cedex 05, France. Correspondence and requests for materials should be addressed to D.L. (email: david.leboeuf@lncmi.cnrs.fr) or to B.F. (email: benoit.fauque@espci.fr)

A magnetic field can induce unusual electronic ground states, such as the quantum Hall effect for a two-dimensional (2D) electron gas. In the limit where only the $n = 0$ Landau level is populated (the so-called quantum limit), electron interactions are responsible for the appearance of a variety of many-body ground states such as the fractional quantum Hall effect[1]. In contrast to the 2D case, the electrons in the quantum limit of a three-dimensional (3D) gas have the ability to move along the direction of the magnetic field. As a consequence, the energy spectrum of the system becomes analogous to a one-dimensional (1D) spectrum in the vicinity of the quantum limit. A variety of electronic instabilities driven by the electron–electron interactions, which may arise in this context, have been proposed[2–4].

In the early 1980s, one phase transition induced by a magnetic field was discovered in graphite[5] (see refs. [6, 7] for reviews). The field-induced state describes a dome in the temperature—magnetic field phase diagram. The onset field of this phase varies with temperature, and is 34 T at 4.2 K. The dome closes at a temperature independent field of 53 T or so, called the re-entrance field. Despite numerous studies, the nature of the order parameter and the role of electron–electron interactions in this transition is still debated. Due to the variety of degrees of freedom competing for the ground state (orbital, spin, and valley), several instabilities have been proposed over the years: charge[8, 9, 10] and spin[11], density wave (respectively labeled CDW and SDW) and more recently excitonic phases[12, 13]. In this article, we refer to this phase with the generic expression density wave (DW) since the precise nature of this phase has not been yet settled.

With the notable exception of a study of the Nernst coefficient[14], the experimental exploration of these instabilities has been limited to measurements of charge conductivity. A study of the field dependence of magnetization failed to report a convincing signature for these instabilities[12, 15].

Here we present the study of elastic properties of highly oriented pyrolytic graphite (HOPG) with ultrasound measurements. Combined with electrical transport measurements, they lead to a rich phase diagram characterized by a sequence of thermodynamic phase transitions. Four transitions are observed: the onset and disappearance of the DW state, and two additional transitions occurring within the ordered state. The DW state appears through a second order phase transition. At higher fields, a first-order transition within the DW dome is observed, which is reminiscent of lock-in transition of the density modulation, a common feature in CDW systems[16]. Our analysis of the DW onset transition reveals that the electron-phonon interaction should be taken into account in theoretical models in addition to electron-electron interactions. The interaction of electrons with the lattice may favor a DW phase with an in-plane component modulation reminiscent of the CDW state[17, 18] and excitonic state[19] proposed in the case of graphene.

## Results

**Overview**. Early ultrasound experiments on HOPG performed up to 2 T documented quantum oscillations in sound velocity change, $\Delta v/v$ and in sound attenuation change $\Delta\alpha(\omega)$[20, 21]. In the low frequency limit, or static limit, the sound velocity $v_{33}$ measured here, corresponding to a longitudinal mode propagating along the $c$-axis of the hexagonal lattice, is given by $v_{33} = \sqrt{\left(\frac{c_{33}}{\rho}\right)}$, where $\rho$ the mass density and $c_{33}$ the corresponding elastic constant, using Voigt notation. In the static limit, the sound velocity is hence a second derivative of the free-energy, as specific heat or thermal expansion. As such the sound velocity is a thermodynamic probe[22]. At finite frequency dispersion effects do

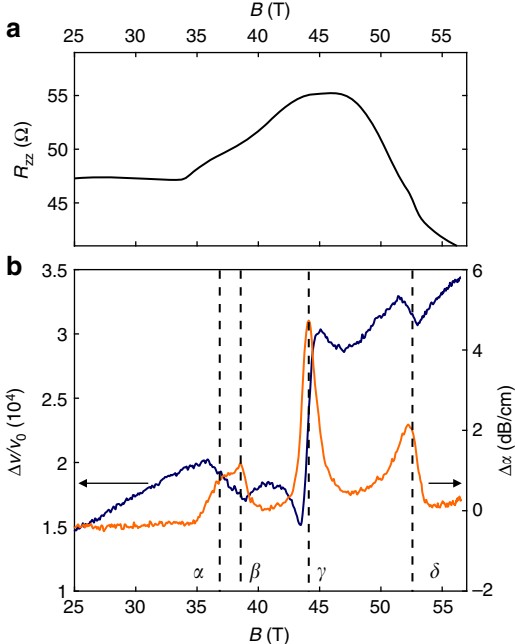

**Fig. 1** High field transport and elastic properties of HOPG at $T = 4.2$ K. **a** $c$-axis magnetoresistance ($R_{zz}$) as a function of magnetic field, $B$, oriented parallel to the $c$-axis in HOPG graphite. **b** elastic properties of HOPG for $B \parallel c$ at 4.2 K: the sound velocity ($\frac{\Delta v}{v_0}$ in orange left $y$-axis) and attenuation ($\Delta\alpha(\omega)$ in blue right $y$-axis) change for the longitudinal acoustic mode $c_{33}$ propagating along the $c$-axis, at a frequency $f = 255$ MHz. The dashed lines indicate the transition field for the different transitions labeled $\alpha$, $\beta$, $\gamma$, and $\delta$

renormalize the sound velocity, because of the coupling of sound waves with internal degrees of freedom. Hence at a second order phase transition we expect two contributions to the sound velocity. First, the thermodynamic discontinuity in the elastic constant, linked to the discontinuities in the isobaric specific heat and thermal expansion through the Ehrenfest relations. The other contribution comes from the coupling to fluctuations and relaxation of the order parameter with characteristic lifetime $\tau$[23]. The static or thermodynamic limit is achieved when $\omega\tau \ll 1$. The distinction between those two contributions will prove useful below.

Figure 1 compares the magnetoresistance with the field dependence of the ultrasound data $\frac{\Delta v}{v_0}$ and $\Delta\alpha(\omega)$ at $T = 4.2$ K and at a frequency $f = 255$ MHz (see Supplementary Note 1 and Supplementary Figs. 1, 2 for detailed information on the ultrasound and transport measurements). At $T = 4.2$ K, a sharp increase of $R_{zz}$ is observed close to 34 T, followed by a subsequent decrease, just before the re-entrance field $B = 53$ T, where the DW disappears. As the temperature increases, the onset shifts toward higher fields. A sequence of anomalies in $\frac{\Delta v}{v_0}$ (jumps and changes of slope) concomitant with peaks in $\Delta\alpha(\omega)$ are clearly visible. Each peak in the attenuation is associated with an anomaly in the sound velocity which signals a phase transition. Different kind of anomalies can be observed in the sound velocity at a phase transition depending on the coupling between the order parameter and the lattice. The comparison between magnetoresistance and ultrasound demonstrates the sensitivity of ultrasound measurements: broad and smooth structures in the resistivity appear as sharp and very well-defined anomalies in ultrasound measurements.

Using magneto-transport data, Yaguchi et al.[24], identified and labeled four phase transitions as $\alpha$, $\beta$, $\gamma$, and $\delta$. Here, we follow their convention to label the ultrasound anomalies as shown in

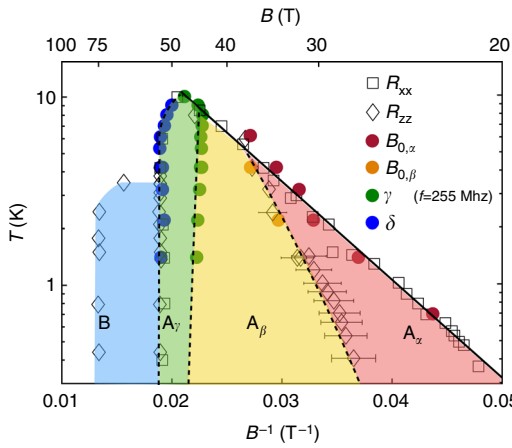

**Fig. 2** $(B^{-1}, \log(T))$ phase diagram of graphite. Two main domes labeled dome A and dome B are found as a function of magnetic field. Dome A consists of a sequence of electronic phases labeled $A_{\alpha,\beta,\gamma,\delta}$ separated by thermodynamic phase transition lines respectively labeled $\alpha, \beta, \gamma,$ and $\delta,$ as shown in Fig. 1. In this work we determined those transition lines with ultrasound measurements (full circles) and transport measurements ($R_{xx}$ and $R_{zz}$ in black open squares and diamonds, respectively). The field scales $B_{0,\alpha}$ and $B_{0,\beta}$ correspond to the ultrasound anomaly at the $\alpha$ and $\beta$ transitions extrapolated at zero frequency using the Landau-Khalatnikov formula (see text). Values reported for the $\gamma$-transition are taken at 255 MHz, and the $\delta$ transition shows a negligible frequency dependence. The black line associated with the $\alpha$ transition corresponds to the behavior expected from a BCS-like description of the DW transition[8]. Dotted lines are guides to the eye for the $\beta, \gamma,$ and $\delta$-transition lines. The boundaries of dome B have been determined previously using $R_{zz}$[25]. The error bars of $B_{0,\beta}$ have been estimated by measurements of $R_{zz}$ in five different samples

Fig. 1. In combination with resistivity measurements (shown in Supplementary Note 2) the ultrasound measurements reported here lead to a detailed phase diagram of the high field states of graphite presented in a semi-log plot in Fig. 2. Before discussing in detail the various phases present in the phase diagram, let us make a number of comments on its general shape. We distinguish two main domes: dome A and dome B. Our ultrasound study is restricted to dome A. When increasing the magnetic field the first phase boundary that is crossed is associated with the $\alpha$ -transition of dome A. Its boundary forms an almost straight line in the $(B^{-1}, \log(T))$ plane, as seen in red in Fig. 2. This means that the critical temperature and magnetic field, respectively $T_0$ and $B_0$, are linked together through a simple formula: $T_0 = T^* \exp(-B^*/B_0)$, where $T^*$ is a temperature scale associated with the Fermi energy, and $B^*$ a field associated with the Landau level dispersion along the field direction[8]. This empirical expression can be understood as a BCS-like formula for critical temperature., where the density of states is proportional to magnetic field, as result of the degeneracy of the Landau levels [6]. Dome A peaks at 50 T, with a maximum $T_c = 10$ K and then ends at a vertical phase boundary at 53 T, shown by blue data points in Fig. 2. The subsidiary phases of dome A are labeled $A_\alpha$, $A_\beta$, and $A_\gamma$. The destruction of phase A leads to another field-induced state, called B. Dome B collapses abruptly at 75 T and has a maximum $T_c = 3.5$ K[25].

Below we describe the information that the ultrasound attenuation and velocity yield regarding each of the phase boundaries associated with dome A. The discussion is centered around the comparison of in-plane and out-of-plane magnetoresistance shown in Fig. 3, and the temperature and frequency dependence of ultrasound properties as a function of magnetic field, that are shown in Figs. 4, 5.

**$\alpha$ and $\beta$ transitions**. The onset of the transitions in the in-plane resistance, $R_{xx}$, and the $c$-axis resistance, $R_{zz}$ is illustrated in Fig. 3 at low temperatures. When the field is increased, $R_{xx}$ first sharply rises, then plateaus and finally increases again. The two successive increases have been attributed to two successive transitions labeled, $\alpha$ and $\beta$-transition[24]. In contrast $R_{zz}$ does not show a plateau, and only increases above a magnetic field close to the $\beta$-transition. As previously noticed[26], the $\alpha$-transition barely affects $R_{zz}$.

The first two peaks observed in the sound attenuation can naturally be attributed to the $\alpha$ and $\beta$ transitions. However, the transition fields in ultrasound measurements differ slightly from transport measurements. As shown in Fig. 5a, b, the magnetic field at which the $\alpha$ and $\beta$ attenuation peaks occur at a certain temperature actually depends on the frequency of the sound wave. The magnetic field positions of the attenuation peak at the $\alpha$ transition is plotted as a function of frequency, for different temperatures ranging from 0.7 to 6.2 K, and shown in Fig. 5c. Each isotherm is well described by a simple formula:

$$f = f_0 \left( \frac{B}{B_0} - 1 \right). \tag{1}$$

Equation (1) is characteristic of an order parameter relaxation process described by the so-called Landau–Khalatnikov (LK) theory[27, 28] first applied to sound propagation just below the $\lambda$-transition in helium-4[29]. The LK theory has been successfully applied since then to various kinds of transitions ranging from liquid/gas[28], nematic/smectic[30], ferroelectric[31], and ferromagnetic transitions[32] (see ref. [28] for a review). The observation of such a relaxation mechanism is a first evidence that a static, long-range, 3D order parameter appears at the $\alpha$-transition in graphite. Below the critical temperature $T_c$, when the sound wave frequency $f$ matches the order parameter relaxation rate $1/\tau_0$ so that $2\pi f \tau_0 = 1$, the energy absorption is the highest and an excess attenuation is expected just below $T_c$. In the case of the $\lambda$-transition, as the temperature gets closer to $T_\lambda$, $\tau$ is diverging as $\frac{\tau_0}{1-T/T_\lambda}$ and the maximum in ultrasound absorption shifts further away from $T_\lambda$ as the frequency increases. Translating this theory to a field-induced phase, the temperature is replaced by the magnetic field and the transition occurs above the critical field $B_0$ leading to Eq. (1). We report on Fig. 2 the temperature dependence of $B_0(T)$ for the $\alpha$ and $\beta$ transitions. The comparison between the ultrasound and transport measurements shows that $B_{0,\alpha}$ matches with the first anomaly in $R_{xx}$ while $B_{0,\beta}$ matches the second anomalies in $R_{xx}$ (concomitant with the large increase in $R_{zz}$).

We note that the characteristic order parameter relaxation rate $1/\tau_0 \approx 8 \times 10^8$ Hz found here at $B_\alpha = 36.1$ T (Supplementary Note 3) is small compared to a magnetic transition such as in Ni[32] where $1/\tau_0 \approx 10^{14}$ Hz but it is large in comparison with the CDW phase of NbSe$_2$ where $1/\tau_0 \approx 10^3$ Hz due to the coupling between the CDW and discommensuration domains[33]. As discussed in Supplementary Note 3, this intermediate relaxation time can be either the result of the coupling of the DW with the lattice or with collective excitations of the DW condensate. Note that in this experiment the field is oriented along the $c$-axis. It would be interesting to study the angular dependence of $f_0$ and compare it with that of $B_0$[13].

In Fig. 5b, we see that the location of the attenuation maximum associated with the $\beta$ transition has a similar frequency dependence to that of the $\alpha$ transition. This indicates that the $\beta$ transition, which does not show hysteresis within our resolution, is a second order transition. However, Fig. 5a shows that the attenuation peak for the $\beta$ transition increases more rapidly than that of the $\alpha$ transition as a function of frequency. Different behaviors at the two transitions are also found in the sound

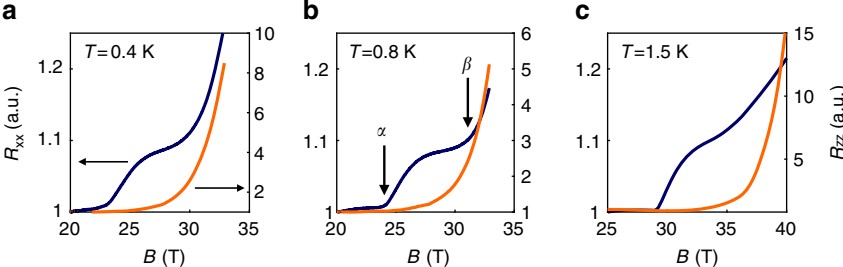

**Fig. 3** In-plane and $c$-axis magnetoresistance in Kish graphite. In-plane ($R_{xx}$ in blue) and $c$-axis ($R_{zz}$ in orange) data obtained at **a** 0.4 K, **b** 0.8 K, and **c** 1.5 K. The magnetic field is oriented along the $c$-axis. For clarity the curves are normalized to their values at $B = 20$ T for **a**, **b** and $B = 25$ T for **c**

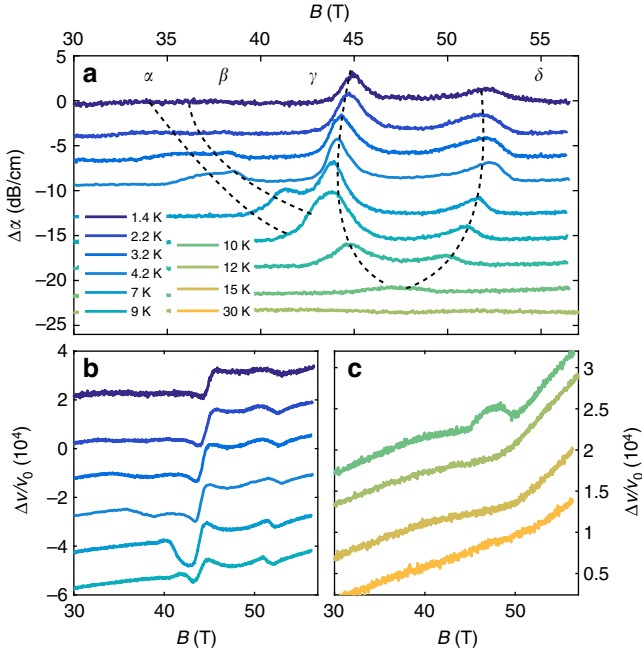

**Fig. 4** Temperature dependence of the elastic properties of graphite. **a** Attenuation as a function of $B$ for $f = 255$ MHz and temperature ranging from 1.4 to 12 K. The dashed lines are eye-guiding to illustrate the temperature dependence of the $\alpha$, $\beta$, $\gamma$, and $\delta$-transitions. The temperature dependence of $\frac{\Delta v}{v_0}$ is shown in **b**, **c** from 1.4 to 9 K and from 10 to 20 K, respectively. Full data set is reported in Supplementary Fig 1.

velocity. In Fig. 1, one observes that the $\alpha$ attenuation peak coincides with the mid-point of a negative step-like anomaly in the sound velocity. Such an anomaly in the elastic constant is another evidence for the occurrence of a phase transition. It occurs at a second order phase transition, when the square of the order parameter couples linearly with the strain in a Landau theory[22]. On the other hand, the ($\beta$)-transition shows up as a change of slope in the sound velocity. The absence of a discontinuity can be either due to the limitation in the experimental resolution or an indication that this transition involves a different order parameter- strain coupling.

The amplitude of the jump is best measured at low frequency in the static or thermodynamic limit where $2\pi f \tau_0 \ll 1$, and where dispersion effects are negligible. We find for the $\alpha$-transition $\frac{\Delta v}{v_0} = \frac{\Delta c_{33}}{2c_{33}(B=0)} = -3 \times 10^{-5} \pm 2 \times 10^{-5}$ at 4.2 K and for $f = 36$ MHz (Supplementary Fig. 1). In the discussion section we will examine the amplitude of this jump. We note that an anomaly in the magnetostriction along the $c$-axis should in principle also exist, but, as explained in Supplementary Note 4, we were not able to

resolve it in our measurement of $L_{33}$, the dimension of the sample along the $c$-axis. For $T < 4.2$ K the anomalies in the sound velocity at the $\alpha$ and $\beta$ transition loses amplitude as temperature is decreased as shown in Fig. 4b. This is due to the fact that as the critical temperature for the transitions decreases, the thermodynamic contribution decreases through reduction of the specific heat.

**$\gamma$-transition**. The $\gamma$-transition differs from other transitions regarding its temperature and frequency dependence. In contrast to the $\alpha$ and $\beta$-transition, the $\gamma$-transition is almost temperature independent, much like the $\delta$-transition (Fig. 4a, b). But its frequency dependence (reported on Fig. 5b) differs from the $\alpha$, $\beta$, and $\delta$-transitions. As the field increases we observe a hardening at the $\gamma$-transition i.e., an opposite behavior compared to the $\alpha$ and $\delta$-transitions. The $\gamma$-transition differs as well in its signature in transport properties. As seen in Fig. 1, the $\gamma$-transition has, by far, the largest effect on the elastic properties of graphite, while its signature in $R_{zz}$ is small compared to the other transitions. Nonetheless, former in-plane magnetoresistance studies[9, 24] identified temperature independent anomalies close to $B \approx 45$ T (where $R_{xx}$ changes slope) and have been associated with a theoretically predicted lock-in transition of a CDW order along the $c$-axis[9]. On the other hand, close to the $\gamma$ transition $R_{zz}$ has a maximum which has been recently linked with the formation of an excitonic phase[13]. Our observation of a large change in the elastic properties inside dome A supports the lock-in scenario of ref. [9] implying a field dependence of the wave vector modulation of the DW. As in the case of 2H-TaSe$_2$, the lock-in transition is almost invisible through transport measurements, but gives the most dramatic effect in elastics properties[33]. Furthermore, by comparing $\frac{\Delta v(B)}{v_0}$ measured in field-up and field-down sweeps (shown in Fig. 5d), we find that the $\gamma$-transition is characterized by an hysteresis loop, which suggests that it is of a first-order nature, contrary to the $\alpha$ transition. Note however that we did not reach the static limit for this transition. Alternatively, the $\gamma$-transition could be of second order character with the hysteresis loop originating from other effects such as the pinning of the CDW.

**$\delta$-transition**. In contrast to the three other transitions discussed so far, the $\delta$-transition is almost both temperature and frequency independent. A weak hysteresis loop is found in the sound velocity at the $\delta$ transition, as shown in Fig. 5d, indicating it has a first-order character. The $\delta$-transition corresponds to the re-entrance field, as determined by transport measurements. This re-entrance has been theoretically attributed to Landau levels (LL) depopulation[34]. We studied the elastic properties above the maximum $T_c$ of dome A and the data are shown in Fig. 4c. At 10 K, the field-induced state arises on top of a local softening or

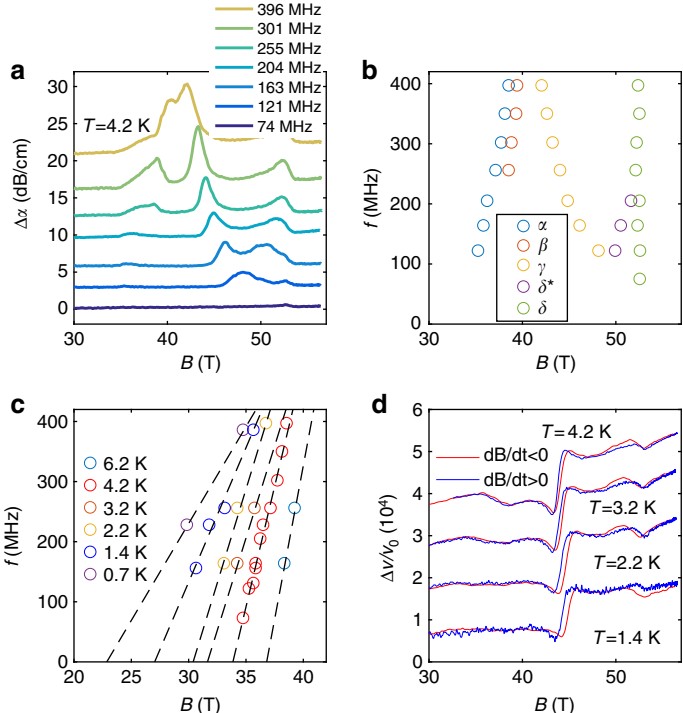

**Fig. 5** Frequency dependence of the sound attenuation. **a** Field dependence of the sound attenuation ($\Delta\alpha$) at $T = 4.2$ K for sound frequencies ranging from 74 to 396 MHz **b** Frequency—magnetic field phase diagram of the peak position shown in **a**. **c** Position of the first attenuation maximum as function of the frequency for temperatures ranging from 6.2 down to 0.7 K. The dashed lines are linear fits to Eq. (1). **d** Sound velocity hysteresis at $\gamma$ and $\delta$ transitions. The red and blue curves correspond to the field-down and field-up sweeps at $f = 255$ MHz, respectively. No hysteresis is observed for the $\alpha$ and $\beta$ transition

minimum in the sound velocity, which is observed even above at 12 and 15 K, i.e., above the maximum $T_c$. This softening is similar to a LL crossing in the sound velocity, as observed at lower fields (Supplementary Note 1). The asymmetric shape of the $\delta$ attenuation peak in the field-induced state, characteristic of a LL crossing, further supports this interpretation. The LL depopulation observed here removes the density of states necessary for the DW state to condense which consequently collapses abruptly at the LL crossing. The $\delta$ transition thus corresponds to the disappearance of the DW state at a first-order phase transition, caused by a LL crossing. We note that below 200 MHz, an additional structure, labeled $\delta^\star$ in Fig. 5b, appears and merges at 100 MHz with the $\gamma$-transition and could be associated with the disappearance of either the $\alpha$ or $\beta$ phase.

## Discussion

In the case of graphite it is generally assumed that the DW forms as a result of a nesting process along the magnetic field direction driven by electron–electron interactions[8]. In that picture the DW modulation runs along the magnetic field axis with a modulation vector that evolves as a function of magnetic field[9]. Here we show that electron–electron interactions are indeed at work. According to theory, in the absence of electron-electron interactions, the depopulation of the LL $(0, +)$ and $(-1, -)$ is expected to occur at a magnetic field significantly larger than 55 T[11]. Once the exchange and correlation effects are included, the bare LL spectrum is renormalized and the depopulation of the LL $(0, +)$ and $(-1, -)$ occurs almost simultaneously at around 52–54 T[9, 11], close to the softening observed in $\Delta v/v_0$ at 12 and 15 K. According to our measurements, the critical temperature is the highest in the vicinity of a LL depopulation, a limit where the interactions have the strongest influence[35].

The amplitude of the sound velocity jump at the $\alpha$-transition is an order of magnitude smaller than the one found at the SDW

transition of $PF_6$[36] and two orders of magnitude smaller than the one found at the incommensurate CDW state of $2H$-$TaSe_2$[33] but it is surprisingly high for a system with a carrier concentration four orders of magnitude smaller than these metals. As discussed in Supplementary Note 5, the large amplitude of the sound velocity anomaly measured at the $\alpha$-transition suggests that, in the formalism of the Ginzburg–Landau theory, the order parameter is coupled to a lattice strain along the $c$-axis. This unexpected coupling raises several questions and calls for further works. What it is the amplitude of the specific heat jump at the DW transitions and the respective contribution of electrons and phonons? Could the electron-phonon interaction also induce an in-plane lattice deformation along with the $c$-axis nesting process as suggested by early theoretical works[37, 38]? Could an in plane lattice deformation explain why the $\alpha$-transition appears in $R_{xx}$ but only weakly in $R_{zz}$ as shown in Fig. 3?

In conclusion, we show the first thermodynamic signatures of the electronic instabilities induced by a magnetic field in graphite. Our data show an experimental evidence for a Landau level crossing at $B \approx 50$ T, observed for temperatures above the onset temperature of the field-induced state. While the wave vector direction of the DW state remains to be determined, our thermodynamic analysis indicate that the order parameter is coupled to a $c$-axis strain, suggesting that the interaction between quasiparticles and the lattice is at play and should be accounted for in theoretical models.

## Methods

**Samples and experimental characteristics.** The ultrasound measurement have been performed on Highly Oriented Pyrolytic Graphite (HOPG) samples grade ZYA that were purchased from Momentive Performance Materials. Longitudinal ultrasonic waves were generated using commercial LiNbO₃ 36° Y-cut transducers glued on a fresh cleaved surface. The magnetic field was aligned along the $c$-axis. Experiments have been conducted in both static field at the HMFL, Nijmegen up to 37.5 T, and pulsed field at the LNCMI-Toulouse up to 58 T. In both case standard

pulse-echo technique was used to determine the change in the sound velocity and attenuation. In total four different samples were studied at different stage of this project. These measurements have been completed with additional magnetostriction and transport measurements on the same HOPG samples and on Kish graphite samples (see Supplementary Notes 2, 4 for more details).[10]

**Data availibilty**. All relevant data are available from the corresponding authors.

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

## Acknowledgements

This work is supported by the Agence Nationale de Recherche as a part of the QUANTUMLIMIT project, and as part of the UNESCOS project (contract ANR-14-CE05-0007), by a grant attributed by the Ile de France regional council, by the Laboratoire d'excellence LANEF in Grenoble (ANR-10-LABX-51-01) and by Université Grenoble-Alpes (SMIng—AGIR). We acknowledge support from the LNCMI and the HFML which are both members of the European Magnetic Field Laboratory. B.F. acknowledges support from Jeunes Equipes de l'Institut de Physique du Collège de France (JEIP). We thank P. Littlewood, P. Monceau, J-Y Prieur and M. Saint-Paul for stimulating discussions.

## Author contributions

D.L. and B.F. planned the experiments. D.L., B.F., M.F. and S.W. conducted the ultrasound experiments and D.L. and B.F did the data analysis. C.W.R., G.S., W.T. and B.F. conducted the transport measurements and C.W.R. and B.F. did the data analysis. R.K. designed the magnetostriction set-up and M.B., S.W. conducted the c-axis magnetostriction measurements and S.W. did the analyze. D.L., K.B. and B.F. wrote the paper with the input of all the authors.

## Additional information

**Competing interests:** The authors declare no competing financial interests.

