## [Peer Review File · Nature Communications]

Reviewers' comments:

Reviewer #1 (Remarks to the Author):

The paper reports high-field pulsed echo ultrasound measurements in graphite. These are the first ultrasound measurements of this kind in graphite in fields up to 60T -- it identifies the ultrasonic signature of the known high-field phase diagram of graphite and makes an attempt to identify the different dynamic behaviour associated with each phase boundary.

As such the results of this study are important and timely and I recommend this paper for publication without delay.

I would recommend though that authors proof-read the text before publication -- e.g. " $(f/2\pi)\tau=1$ " in the middle of the paragraph following Eq. 1. ; also some cleanup in the equations in the text immediately following Eq. 2.

I have two general comments.

Regarding the discussion of the "gamma" transition on page 4, I want to point out that velocity experiences an upturn (as field is increased) -- in contrast to downturn for "alpha" transition. Figure 1 suggests that the magnitude of the peak in attenuation (considered as $\text{inverse-Q} \sim 2\pi (\alpha c)/\omega$) corresponds in magnitude with the relative change of velocity $\Delta V/V$ across the upturn. This suggests that the upturn in velocity and the attenuation feature are Kramers-Kronig-related to each other. This is also supported by the opposite trend of the frequency dependence exhibited by "gamma" peak as opposed to "alpha" peak (Figure 5b). This suggests fluctuation (rather than thermodynamic) origin of the gamma-attenuation-peak where the velocity upturn is a mere consequence of strong slowing down of dynamics via the analyticity relations between attenuation and velocity change, inherent for the ultrasound measurements. Dynamic origin of the feature associated with "gamma" seem to be also consistent with electric transport studies in the same field range (in particular, their angular dependence).

Another general comment is on the discussion section on page 5. I suggest the authors make clearer (and more grounded in their data) the discussion of the "importance of the lattice exposed here". In particular, I did not get an impression that the orders-of-magnitude mismatch discussed in relation to Ehrenfest relation, Eq. 2, is well established because it relies on BCS-like relation of the jump in C/T to the electronic density of states -- not expected to hold in this low-electron-density system.

It is not clear from the discussion in the paper how "lattice" helps to resolve this mismatch. In this relation, it is also not clear what is meant by "the role of the lattice can be relevant to understand..." in the second-to-last paragraph and "dominant lattice response occurs..." in the last paragraph.

Finally, I want to point out that I don't see how the speculative discussion in the end of the paper of possible symmetry changes, the q-vectors associated with each phase boundary, and their possible

microscopic origins (for which the existing geometry of this measurement cannot provide much evidence) -- how it helps the main message of the paper. The measurements reported in the paper are important and challenging in their own right.

Reviewer #2 (Remarks to the Author):

The authors present an impressive study of the high-magnetic field properties of graphite comparing magnetoresistance and ultrasound measurements to establish a detailed B-T phase diagram.

By a thermodynamic analysis of the change of the sound velocity on the alpha-phase transition, the authors conclude that electron-electron interaction alone is not sufficient to account for this transition. The existence of a further order parameter related to the lattice is evoked to account for this discrepancy. The thorough discussion of the involved approximations and estimates in the supplementary material is very clear and concise.

In my opinion, this manuscript provides convincing evidence for the importance of electron-lattice coupling in the ground state of graphite, which I consider an important contribution to the field and of interest for a wider range of readers. Therefore, I recommend to accept this manuscript for publication in Nature Communications.

Reviewer #3 (Remarks to the Author):

The density wave instabilities induced by strong magnetic fields in graphite, in the 1D dispersing Landau levels, is a topic with a long history. The problem is of fundamental importance and is as yet unsolved. It is attracting renewed interest, albeit from a small number of groups, with a number of recent papers referred to in this manuscript, some of which are only on the arXiv. Probably this renewed interest in graphite is partly fuelled by the obvious connections with research on graphene.

This manuscript comes from a group active in the study of this system, with a relatively recent paper Fauque et al (2013), ref [22] of the manuscript.

The various studies of magnetotransport in this system, using pulsed magnetic fields, provide evidence for the onset of DW instabilities in the remaining four (electron/hole spin split) Landau Levels with one dimensional dispersion. These states are quenched as Landau levels empty on increasing the field. As the manuscript states, the nature of the DW states is not settled. This remains one focus of current research.

This manuscript reports new measurements, using ultrasound as the experimental probe, in magnetic fields up to 58T. These measurements complement magnetotransport in an important way

due to sensitivity to renormalisation of elastic coefficients concomitant with the DW instabilities in the electronic system.

In general terms the results here correspond closely to and validate signatures from magnetotransport, and in some cases confirm interpretations suggested in previous work.

What is most important is that these ultrasound measurements reveal relatively distinct and sharp features. They are likely to be helpful in establishing the nature of the DW instabilities. However the authors are somewhat unclear/reticent in the discussion section on what is learned of the nature of the DW instabilities from these measurements.

My reservations about the manuscript in its current form revolve around:

- (i) the claim that a “thermodynamic probe” is being used to determine the phase diagram.
- (ii) the way the present result is contextualized, and discrimination between what is truly novel in the result and its interpretation and what is not.
- (iii) proper cross-referencing with published results reflecting (ii)
- (iv) issues of rigor/clarity/quality in the scientific presentation, and caveats where matters are just uncertain.

Nevertheless the novelty of these measurements, their timeliness, and the importance of the topic definitely merits publication in Nature Communications if the above issues are properly addressed. See also concluding remarks.

In more detail:

1. Ultrasound measurements.

(i) The distinction between attenuation as a transport property and sound velocity as a thermodynamic probe (page 2 column 1) is unwarranted and artificial. Ultrasound probes the viscoelastic properties of the medium, and attenuation and velocity are the imaginary and real components of the response function. The relaxation time enters both. There is no sharp “step” in sound velocity, observed or expected, despite several references to a “step” or “jump; it is rounded by relaxation.

(ii) The explanation of the procedure to determine the critical field of the field induced transition should be clarified and improved. The Landau-Khalatnikov theory [this paper should be cited] applies to a thermally driven second order phase transition, and when developed for superfluid 4He, found a divergence of the relaxation time at T_c . If this approach is extended to different systems (the key is the diverging relaxation time at T_c), finite frequency measurements can be used to infer the thermodynamic transition temperature. Thus the attenuation maximum, at which $\omega\tau=1$, determines τ ; then measurements of that maximum as a function of ω determine T_c .

[I suggest it would be helpful to cite/discuss a few more systems where ultrasonic attenuation has been used to identify phase transitions in this way using LK theory, to justify/establish its general applicability]. That said, it seems to work here.

However, here the tuning parameter is magnetic field B . Thus the scaling $1/\tau$ with $B-B_0$, in my opinion, actually has the status of a reasonable phenomenological assumption, consistent with experiment.

(iii) The treatment of the frequency dependence is nice, and is used to determine the transition field. However I do not believe that this can be called a thermodynamic measurement, because B_0 is determined from the frequency dependence of τ . We are in the domain of dynamical critical phenomena [Hohenberg and Halperin, Rev. Mod. Phys. 49, 435 (1977)].

(iv) By the way, neither should this technique be described as ultrasound spectroscopy, which implies transitions between states due to phonon adsorption.

(v) Not remarked upon in Fig 4. The attenuation maxima at α and β appear to be washed out as the temperature is lowered. An improved discussion of the magnitude of the observed anomalies in both sound velocity and attenuation, and their temperature dependence, is desirable. See also comments below.

More speculatively one wonders: At low T this transition, driven by B , is a quantum phase transition. Might quantum fluctuations play a role?

Order of phase transitions

The abstract refers to sequence of second and first order phase transitions. It would be good to be clearer about the order of transitions, and supporting evidence throughout paper. Thus (page 3 column 2 last para), it is clear that the α transition is second order, the analysis is predicated on this and the data is consistent with the model. However, there needs to be a clearer statement about β [possibly "order of transition not determined"?], although later in manuscript (p4 column 2) it is claimed to be not first order. Evidence? At (page 3 column 2 last para) the narrative needs to better match observations: I can't see "steps" or "jumps" in data, at α and β .

γ transition

The largest anomaly, with a distinct maximum in attenuation in the ultrasonic response, occurs at a field between 40 and 50T. This is attributed to a lockin transition in DW order. A feature in magnetoresistance was identified in ref 21 at 42-43T. Ref 9 finds a feature in the temperature range 2-3K at 47T, at which the nesting vector of Landau levels calculated theoretically predicts a lockin transition. It seems that the proposal of a lockin transition is attributable to ref 9; the manuscript should be clearer about this.

The data presented in this manuscript is worthy of expanded discussion and clarification, to better explore the evidence for a lockin transition from ultrasound. For example:

(i) In Fig. 5b the gamma feature (maximum in sound attenuation) shows a strong frequency dependence in field (data at 4.2K). The text should draw attention to this observation, and comment on it.

[After all, the frequency dependence of the alpha feature was subjected to careful analysis].

(ii) This matters, in part, because in plotting the phase diagram of fig. 2 the gamma feature at 285 MHz is selected for plotting. This is clearly indicated in the figure, but the significance (arbitrariness?)

of this choice only becomes apparent on study of Fig. 5b.

(iii) At 4.2 K the gamma feature is starting to merge with beta feature. A frequency dependence of the gamma feature at the lowest temperature would be most informative, and less ambiguous, if available.

(iv) Is the sign of the change in sound velocity consistent with a lock-in transition?

(v) Since the nesting vector is a function of magnetic field you'd expect commensuration only to occur over a narrow field range. If so why does the sound velocity increase (more or less) monotonically with increasing field.

(vi) Might be pinning of the commensurate DW be potentially responsible for the observed hysteresis in gamma, which the manuscript attributes to first order nature of gamma transition? [Although you would expect a lockin transition to be first order].

Discussion section

The authors state in this section:

“Here we show that electron-electron interactions are indeed at work”.

“According to our measurements the DW state forms in the vicinity of a LL depopulation, a limit where the interactions have the strongest evidence”.

And earlier (p2 column 2)

“These anomalies lead to the new phase diagram presented.....in Fig.2”

This text creates the wrong impression of the new knowledge added by the reported ultrasound data.

The ultrasound confirms inferences from magnetoresistance measurements and the phase diagram shown in fig 2 cannot be described as new. The dome B, reported by this group in an earlier paper [ref 22] was an important discovery, although misinterpreted in that work in terms of the Landau levels involved [ref. 9].

Electronic DWs and the lattice

The remaining part of the discussion, expanded in the SM, is interesting but not totally convincing. It concerns a discussion of the magnitude of signature in sound velocity at the α transition.

(i) I think it would be good, in the first instance, to concentrate on the data, in particular better documenting the frequency and temperature dependence of both the sound velocity and attenuation at the α transition. Analysis by a simple viscoelastic model, would hopefully yield the “thermodynamic” jump in sound velocity, and its temperature dependence. This would better pin down the experimental information from ultrasonics before embarking on playing with the thermodynamic relations.

(i) Thermodynamic relations

Internal thermodynamic consistency is clearly an important check and source of insight. The problem is better pin down the related parameters (see also (i) above). I suggest that more care should be taken with this discussion, with more detail added on provenance of input data/assumptions, and associated assumptions. Thus:

(i) The data of Iye on the pressure dependence of T_c of the alpha transition, shows that dT_c/dP is

somewhat field dependent, in the relevant field range. This can also be quantified from eq 4 of that paper (taking the zero pressure limit of the derivative).

(ii) There seems to be limited knowledge of the heat capacity of graphite in a magnetic field. The cited paper, ref. 31 (a paper which itself has previously no recorded citations), states that the electronic heat capacity is strongly field dependent, and shows an unexplained drop above 8T. So I question the justification of the number used, which uses the low field Sommerfeld coefficient! At least these uncertainties should be explicitly referred to.

(iii) The strength of the claim that the signature in sound velocity at the alpha transition is inconsistent with thermodynamic analysis and points to new physics, in the light of these uncertainties, should be evaluated.

The discussion of Iye et al. is all with respect to the SWMc band structure, which does not account for the field dependence of the Landau level structure of graphite due to the neglect of strong correlation effects.

Given the highly anisotropic elastic and electronic properties of graphite, the elastic properties may respond as a witness to transitions driven by electron correlation effects (in the presence of the field dependent Landau level structure), and be renormalized by those correlations. The emphasis on the "importance of the lattice" may be misplaced.

Leaving aside these reservations I would urge the authors in any case to take another look at the drafting of the final two paragraphs particularly the penultimate paragraph. [minor point: The word "pattern" appears four times in five lines.] The first and last sentence of this paragraph are very vague.

A sharper discussion of how the experimental observations of this paper lead to the proposal of an in-plane DW would be of great interest and enhance the paper.

Minor detail:

The authors must explicitly state the direction of sound propagation (along c-axis) in the main manuscript. This is neither stated in the main body of the paper, nor the methods section. I suggest it appears in the former.

Conclusion

In conclusion, and to repeat aspects of the assessment made earlier in this review, the data presented here represent a new approach towards understanding the nature of field-induced electronic states in graphite. The method also holds future promise, an obvious example being extension to higher fields ("dome B2").

In my opinion these results should be published in Nature Communications. However work is required to improve the current version of the manuscript, and I hope that the comments above prove helpful.

Answers to referee 1 :

We thank referee 1 for her/his report and for her/his recommendation for publication of our results. We report below our answers to her/his questions and the modification of our manuscript based on his comments.

[1] I would recommend though that authors proof-read the text before publication – e.g. "($f/2\pi$) $\tau=1$ " in the middle of the paragraph following Eq. 1. ; also some cleanup in the equations in the text immediately following Eq. 2.

Response : The text has now been carefully proof-read, several changes were made, that are listed below.

[2] Regarding the discussion of the "gamma" transition on page 4, I want to point out that velocity experiences an upturn (as field is increased) – in contrast to downturn for "alpha" transition. Figure 1 suggests that the magnitude of the peak in attenuation (considered as inverse-Q $2\pi(\alpha c)/\omega$) corresponds in magnitude with the relative change of velocity $\delta V/V$ across the upturn. This suggests that the upturn in velocity and the attenuation feature are Kramers-Kronig-related to each other. This is also supported by the opposite trend of the frequency dependence exhibited by "gamma" peak as opposed to "alpha" peak (Figure 5b). This suggests fluctuation (rather than thermodynamic) origin of the γ -attenuation-peak where the velocity upturn is a mere consequence of strong slowing down of dynamics via the analyticity relations between attenuation and velocity change, inherent for the ultrasound measurements. Dynamic origin of the feature associated with "gamma" seem to be also consistent with electric transport studies in the same field range (in particular, their angular dependence).

Response : We agree with referee 1 that the γ transition is different from the other transition and could be attributed to a fluctuating phenomenon rather than a transition to a static phase. Indeed our measurements are performed at finite frequency, and can couple to fluctuating degrees of freedom. To distinguish the effect of fluctuation from the effect of static, thermodynamic, phenomena, one needs to carry on experiment below the cutoff frequency of the fluctuation spectra. In this static limit, the thermodynamic contribution to the elastic constant can be isolated. Within our frequency window such regime has only been reached for the alpha transition (see SM Fig. 1 d) and not for the γ transition. However, resistivity measurement (done at frequencies as low as 30kHz) have, as well, identified a γ transition (see for example [9,23]) and thus point out for a static origin of this transition. Also the γ transition is first order and corresponds to the largest elastic constant change. In the context of a CDW transition those facts are naturally indicating a lock-in transition of the CDW state which should be a static phase.

[3] Another general comment is on the discussion section on page 5. I suggest the authors make clearer (and more grounded in their data) the discussion of the "importance of the lattice exposed here". In particular, I did not get an impression that the orders-of-magnitude mismatch discussed in relation to Ehrenfest relation, Eq. 2, is well established because it relies on BCS-like relation of the jump in C/T to the electronic density of states – not expected to hold in this low-electron-density system.

It is not clear from the discussion in the paper how "lattice" helps to resolve this mismatch. In this relation, it is also not clear what is meant by "the role of the lattice can be relevant to understand..." in the second-to-last paragraph and "dominant lattice response occurs." in the last paragraph.

We do agree with the referee that our estimation is only putative, since we currently don't know the amplitude of the jump in the specific heat at the transition. Yet, the magnitude of the available electronic entropy is known. As the referee recalls this is a low-density system and therefore the electronic entropy is very dilute. The thermodynamic restrictions imposed by equation 2 imply that a jump in sound velocity requires either a large jump in specific heat (three orders of magnitude larger than what is available in electronic reservoir) or an extreme equality between the dependence of the critical temperature on pressure and on strain. Either of the two possibilities imply a significant role for lattice.

There are two ways to involve lattice in a CDW transition. First, it is possible that a phase transition involving the electronic degrees of freedom coexists with a structural transition. Second, strong coupling to

the lattice may cause a high strain dependence of T_c in order to produce the large anomaly observed here in the sound velocity. In many CDW systems, lattice distortion accompanies the phase transition. FFermi surface effects are often as important as electron-phonon coupling, and elastic energy. In the case of graphite, the lattice has been left out of theories produced so far. Based on comments by referees 1 and 3, we have reformulated our statement on the possible role of the electron-phonon interaction.

We have changed and simplified the last paragraph of p5. It reads now as follows:

A previous experiment has quantified the magnitude of $\frac{dT_c}{dP}$ ($\frac{dT_c}{dP} \approx 0.3 \text{ K.GPa}^{-1}$) [36]. Here, we detect a jump in sound velocity as large as $\frac{\Delta v}{v_0} \approx 5.10^{-5}$. Assuming $\frac{dT_c}{dP} \approx \frac{dT_c}{dP_3}$, the expected jump of specific heat by Equation 2 is three orders of magnitude larger than the experimentally-resolved electronic specific heat ($\gamma_{el} \approx 14 \mu\text{J.K}^{-2}.\text{mol}^{-1}$ [37]). This discrepancy can be either the result of a strong anisotropy in the strain dependence of T_c or a jump in the specific heat provided by lattice. This is discussed in more detail in section F of the SM.

Until now, the role of the electron-phonon interaction in the formation of the field induce states of graphite has not been considered. Yet it has been shown that in a Hartree-Fock approximation a DW state with a modulation vector parallel to the magnetic field can be accompanied by an in-plane modulation pinned down by the electron-phonon interaction [38,39]. We thus speculate that the α -transition corresponds to an electron-phonon assisted in-plane DW state concomitant with a lattice distortion, which would only affect R_{xx} . At the β -transition, the c-axis conductance is drastically reduced which could be due, for example, to a change in the c-axis pattern of the in-plane DW phase.

[4] Finally, I want to point out that I don't see how the speculative discussion in the end of the paper of possible symmetry changes, the q-vectors associated with each phase boundary, and their possible microscopic origins (for which the existing geometry of this measurement cannot provide much evidence) – how it helps the main message of the paper. The measurements reported in the paper are important and challenging in their own right.

Response: We do agree with the referee that this last paragraph is not essential for the main message of the paper and we are glad to read that the referee found our results important in their own right. Following the referee comments we have simplified this discussion (see previous answer). However, we want to underline that our measurement of R_{xx} and R_{zz} at low temperature (figure 3 of the manuscript) shows that these instabilities first manifest in the in-plane resistance and then in the c-axis transport measurement. It can be naturally explained by invoking the formation of in-plane DW state at the alpha transition. We believed that it is an important message since to our knowledge such scenario has never been considered so far and may motivate experimental or theoretical works to explore the role of the electron phonon interaction in the quantum limit of graphite.

Answers to referee 3 :

We thank referee 3 for her/his careful reading and detailed report on our manuscript. We are pleased that he found our result important. Below we report our answers to her/his very interesting and useful questions/comments and the modification of our manuscript based on them.

Ultrasound measurements : [1] The distinction between attenuation as a transport property and sound velocity as a thermodynamic probe (page 2 column 1) is unwarranted and artificial. Ultrasound probes the viscoelastic properties of the medium, and attenuation and velocity are the imaginary and real components of the response function. The relaxation time enters both. There is no sharp step in sound velocity, observed or expected, despite several references to a step or jump; it is rounded by relaxation.

Following the advice of the referee we have removed the sentence : *While attenuation is a transport quantity associated with phonon lifetime, sound velocity is a thermo- dynamic probe.* We have kept a short discussion on the two distinct contributions which can affect the sound velocity at a phase transition in order to introduce to the reader the thermodynamic analysis done in the section *Discussion* of the text. The first contribution, the static or thermodynamic one, is associated with the apparition of a static order parameter which can change the elastic constant. The second contribution comes from the coupling to fluctuations or relaxation of the order parameter with characteristic frequency F . In common systems, such as ferromagnetic nickel, F is of order 10^{12} - 10^{14} Hz, way above the ultrasound frequency 10^8 MHz. In this regime (where $\omega\tau \ll 1$), the sound velocity around the phase transition is a thermodynamic quantity. An internal check can be done by applying the Ehrenfest relationship, which links the specific heat and the sound velocity together.

In the case of graphite, the fluctuation spectra around the alpha transition has a rather low characteristic frequency 800 MHz at 36.1 T. In this case we need to identify the static regime by checking below which frequency the amplitude of the sound velocity anomaly at the transition do not evolve anymore. This is shown in SM fig 1 d, where we see that below 70 MHz or so the amplitude of the anomaly does not evolve anymore. Hence it can be used to perform a thermodynamic analysis.

[2] I suggest it would be helpful to cite/discuss a few more systems where ultrasonic attenuation has been used to identify phase transitions in this way using LK theory, to justify/establish its general applicability.

Response: We agree and we added the following sentence (p.3 column 2):

The LK theory has been successfully applied since then to various kinds of transitions ranging from liquid/gas[28], nematic/smectic [29], ferroelectric [30] and ferromagnetic transitions [31] (see [28] for a review).

[3] The treatment of the frequency dependence is nice, and is used to determine the transition field. However I do not believe that this can be called a thermodynamic measurement, because B_0 is determined from the frequency dependence of tau. We are in the domain of dynamical critical phenomena [Hohenberg and Halperin, Rev. Mod. Phys. 49, 435 (1977)].

Response: We agree that the determination of B_0 from the extrapolation to zero frequency of the position of the attenuation peak is not a thermodynamic measurement. In that sense the phase diagram of figure 2 is not deduced from a thermodynamic quantity. The title was changed to : *Thermodynamic signatures of the field-induced states of graphite.* We keep the word thermodynamic because, for the reason explained above, at low frequency the sound velocity is a thermodynamic quantity.

[4] By the way, neither should this technique be described as ultrasound spectroscopy, which implies transitions between states due to phonon adsorption.

Response: We changed the sentence *Here we present the study of elastic properties of Highly Oriented Pyrolytic Graphite (HOPG) through ultrasound spectroscopy measurements.* to *Here we present the study of elastic properties of Highly Oriented Pyrolytic Graphite (HOPG) through ultrasound measurements.*

[5] Not remarked upon in Fig 4. The attenuation maxima at α and β appear to be washed out as the

temperature is lowered. An improved discussion of the magnitude of the observed anomalies in both sound velocity and attenuation, and their temperature dependence, is desirable. More speculatively one wonders: At low T this transition, driven by B, is a quantum phase transition. Might quantum fluctuations play a role?

Response: To improve the discussion on the temperature dependence we added on p. 4 column 2:

For $T < 4.2K$ the anomalies in the sound velocity at the α and β transition lose amplitude as temperature is decreased; as shown in Fig. 4b. This is due to the fact that as the critical temperature for the transitions decreases, the thermodynamic contribution decreases through reduction of the specific heat.

Concerning the temperature dependence of the amplitude of the attenuation peaks at the alpha and beta transition, clues can be found in the temperature dependence of the relaxation time. In Figure 1 below we extract f_0 from Fig. 5 c of the main text in order to deduce $\tau_0=1/(2\pi f_0)$. It appears that as T is reduced τ_0 is greatly enhanced. In LK theory excess attenuation goes as $\omega^2\tau/(1 + \omega^2\tau^2)$. If τ increases then the attenuation decreases, and this might explain why the attenuation peak gets smaller as T is lowered.

Finally concerning quantum fluctuations, it is quite interesting to note that τ_0 deduced when the ground state is controlled by a non-thermal parameter (magnetic field) is different (smaller) than τ_0 deduced when the ground state is controlled by temperature. Indeed at 4.2 K, in Fig. 1 below we find $f_0 \approx 2.5$ GHz, whereas from SM Fig. 3 we found $f_0 \approx 0.6$ GHz. This difference might be the signature of the contribution of quantum fluctuations. We added a sentence to highlight this difference in the SM, Section C:

f_0 can also be determined using Fig. 5c of the main text. At 5 K we get $f_0 \approx 3$ GHz, slightly higher than the value determined from the temperature dependence. This indicates that the fluctuation spectrum is different when the ground state of the system is controlled with a non-thermal parameter.

Figure 1 - f_0 (left) and τ_0 (right) as a function of temperature as deduced from Fig. 5 c of the main manuscript.

Order of phase transition. [6] The abstract refers to sequence of second and first order phase transitions. It would be good to be clearer about the order of transitions, and supporting evidence throughout paper. Thus (page 3 column 2 last para), it is clear that the α transition is second order, the analysis is predicated on this and the data is consistent with the model. However, there needs to be a clearer statement about β [possibly order of transition not determined?], although later in manuscript (p4 column 2) it is claimed to be not first order. Evidence? At (page 3 column 2 last para) the narrative needs to better match observations: I cant see steps or jumps in data, at α and β .

Response: The β transition follows the same LK formalism as the α transition, this is why it is believed to be 2nd order. We added the following sentence to clarify the situation (p.4 column 1).

In figure 5, panel b, we see that the location of the attenuation maximum associated with the β transitions have a similar frequency dependence to that of the α transition. This indicates that the β transition, which does not show hysteresis within our resolution, is a second order transition.

At the β -transition we do not observe a step or a jump. We observe a change of slope as stated in the

manuscript. We use the term step or jump for the alpha transition because, at low frequency the anomaly is a step-like feature, as shown in Fig. 1d of the SM. Below we show data at 130 MHz obtained in DC magnetic field up to 37.5 T at 4.2 K. We subtract a phenomenological linear background to the data and the resulting plot shows a well-defined step-like feature.

Figure 2 Left - $\Delta v/v$ as function of the magnetic field for $f=130\text{MHz}$ and $T=4.2\text{K}$. Right - Same data where we subtracted a linear background which highlight the decrease and step like behavior of the sound velocity at the α -transition.

γ -transition : [7] The largest anomaly, with a distinct maximum in attenuation in the ultrasonic response, occurs at a field between 40 and 50T. This is attributed to a lockin transition in DW order. A feature in magnetoresistance was identified in ref 21 at 42-43T. Ref 9 finds a feature in the temperature range 2-3K at 47T, at which the nesting vector of Landau levels calculated theoretically predicts a lockin transition. It seems that the proposal of a lockin transition is attributable to ref 9; the manuscript should be clearer about this.

Response: We rephrased this part in order to make it clearer that the proposal of a lock-in transition was made by Arnold et al. (2014) and that our measurements only provides additional experimental support for this proposal:

p.4 column 2:

Nonetheless, former in-plane magnetoresistance studies[9,23] identified temperature independent anomalies close to $B \approx 45$ T (where R_{xx} changes slope) and have been associated with a theoretically predicted lock-in transition of a CDW order along the c-axis [9]

and

Our observation of a large change in the elastic properties inside the dome A supports the lock-in scenario of Ref.[9].

[8] In Fig. 5b the γ feature (maximum in sound attenuation) shows a strong frequency dependence in field (data at 4.2K). The text should draw attention to this observation, and comment on it. [After all, the frequency dependence of the alpha feature was subjected to careful analysis]. This matters, in part, because in plotting the phase diagram of fig. 2 the gamma feature at 285 MHz is selected for plotting. This is clearly indicated in the figure, but the significance (arbitrariness?) of this choice only becomes apparent on study of Fig. 5b.

Response: In the phase diagram of fig. 2 the γ feature was plotted at 255 MHz because our most complete data set was obtained at 255 MHz (note that in the Fig.2 there was an error, the frequency is 255 MHz not 285 MHz). The gamma transition occurs at lower fields, closer to the onset field for the charge order, when the frequency is increased. This behavior is similar to what is observed at the lock-in transition of the CDW in TaSe₂. In this system the lock-in transition has hysteretic behavior upon cooling

and warming the system, and as frequency is increased, the lock-in transition rapidly moves towards higher temperatures, closer to the onset of the charge order (Barmatz PRB 12 4367 1975).

[9] At 4.2 K the gamma feature is starting to merge with β feature. A frequency dependence of the gamma feature at the lowest temperature would be most informative, and less ambiguous, if available.

Response: We show on figure 3 the field dependence of the sound attenuation at $T=2.2\text{K}$ (the lowest temperature that we have measured in pulsed field) for three frequencies : 163,255 and 395Mhz. As we can see the trend is very close to what we found at $T=4.2\text{K}$ and reported on Fig.5) a) of the manuscript :the γ transition goes to lower magnetic field as the frequency increases and merged with the β -transition.

Figure 3 : Field dependence of the sound attenuation at $T=2.2\text{K}$ and for frequencies $f=63, 255$ and 395Mhz .

[11] Is the sign of the change in sound velocity consistent with a lock-in transition?

Response: The increase of the sound velocity at the lock-in transition is consistent with what is observed in TaSe_2 where the lattice hardens steeply below 90 K (Jericho PRB 22 4907 1980).

[12] Since the nesting vector is a function of magnetic field you would expect commensuration only to occur over a narrow field range. If so why does the sound velocity increase (more or less) monotonically with increasing field.

Response : The referee is right according to the work of F.Arnold et al. [9] the commensurability of the nesting vector of the CDW phase with the lattice parameter only occurs in a narrow range of magnetic field. In this model the CDW phase only form in the Landau levels $(0,\uparrow)$ and $(-1,\downarrow)$ meaning that the two lowest

Landau levels of the electron and hole pocket (respectively $(0, \downarrow)$ and $(-1, \uparrow)$) are not affected by the transition. Thus, these two Landau levels should contribute in the same way after and before the transition where the sound velocity change almost linearly with the magnetic field (as illustrated in figure 2 or Figure 1 of the SM). Currently we are working on a quantitative understanding of this field dependence which is most likely related with the progressive change in the band structure and Fermi energy induced by the magnetic field.

[13] Might be pinning of the commensurate DW be potentially responsible for the observed hysteresis in gamma, which the manuscript attributes to first order nature of gamma transition? [Although you would expect a lockin transition to be first order]

Response: It might be. But would not you expect the transition to occur at lower fields for $dB/dt < 0$ than for $dB/dt > 0$ in that case? Why would pinning be field-induced? Apart from the questions that pinning raises, we are not able to distinguish pinning from a lock-in transition. We added a sentence to mention that possibility. p. 4, column 2, last paragraph: *Alternatively the hysteresis at the γ transition could originate from pinning of the CDW.*

Discussion : [14] The authors state in this section: Here we show that electron-electron interactions are indeed at work. According to our measurements the DW state forms in the vicinity of a LL depopulation, a limit where the interactions have the strongest evidence. And earlier (p2 column 2) These anomalies lead to the new phase diagram presented in Fig.2. This text creates the wrong impression of the new knowledge added by the reported ultrasound data. The ultrasound confirms inferences from magnetoresistance measurements and the phase diagram shown in fig 2 cannot be described as new. The dome B, reported by this group in an earlier paper [ref 22] was an important discovery, although misinterpreted in that work in terms of the Landau levels involved [ref. 9].

Response: Referee 3 makes citations of the submitted manuscript concerning two different facts that are phrased as new facts in the manuscript. First is the phase diagram of Fig. 2. We agree with referee that our phrasing gives the wrong impression that this phase diagram was discovered by the present study. However it is the first time that it is drawn in such a detailed and complete manner. As such we changed the sentence to:

In combination with resistivity measurements (shown in SM) the ultrasound measurements reported here lead to a detailed phase diagram of the high field states of graphite presented in the semi-log plot in Fig. 2.

The second fact discussed here by referee 2 is the evidence for electron-electron interactions and their impact on the Landau level spectrum. While it was shown theoretically that electron interaction renormalized the Landau spectrum and played a role in the stabilization of the DW, it was never evidenced experimentally. Here our sound velocity measurements at a temperature above the onset of DW phase shows that a feature associated with a Landau level crossing does exist at a field in vicinity to the onset field of the DW state. That is why we believe that statements such as *Here we show that electron-electron interactions are indeed at work* and *According to our measurements the DW state forms in the vicinity of a LL depopulation, a limit where the interactions have the strongest influence* are properly phrased.

[15] Internal thermodynamic consistency is clearly an important check and source of insight. The problem is better pin down the related parameters (see also (i) above). I suggest that more care should be taken with this discussion, with more detail added on provenance of input data/assumptions, and associated assumptions. Thus: The data of Iye on the pressure dependence of T_c of the α transition, shows that dT_c/dP is somewhat field dependent, in the relevant field range. This can also be quantified from eq 4 of that paper (taking the zero pressure limit of the derivative).

(ii) There seems to be limited knowledge of the heat capacity of graphite in a magnetic field. The cited paper, ref. 31 (a paper which itself has previously no recorded citations), states that the electronic heat capacity is strongly field dependent, and shows an unexplained drop above 8T. So I question the justification of the number used, which uses the low field Sommerfeld coefficient! At least these uncertainties should be

explicitly referred to.

(iii) The strength of the claim that the signature in sound velocity at the α transition is inconsistent with thermodynamic analysis and points to new physics, in the light of these uncertainties, should be evaluated.

Following the advice of the referee we have changed the discussion on our thermodynamic analysis in the main manuscript and the Supplementary Materiel.

(i) we used the Eq.(4) of Iye and al., PRB 41, 3249 (1990) in the zero pressure limit to evaluate $\frac{dT_c}{dP} = \alpha(1 - B^*/B) * T_c$ where $\alpha = 0.0280.002 \text{ kBar}^{-1}$ (according to the fit of Fig.3 of the work of Iye). The estimation of the jump has be done at $T=4.2\text{K}$ where the critical field is about 30T. As a result : $\frac{dT_c}{dP} = 0.24 \text{ K.GPa}^{-1}$ slightly lower than what we have previously used.

For the point (ii) and (iii) we do agree with referee 3 that there is little knowledge on the specific heat of the electrons at low temperature and high magnetic field in graphite. As the result we can only make estimations using Eq. (2). Nonetheless there is consensus on the zero field value (see for example the works of B. J. C. van der Hoeven, Jr. and P. H. Keesom, Phys. Rev. 130, 1318 (1963) , M.G. Alexander, D.P. Goshorn and D.G. Onn, Phys. Rev. B 22, 4535 (1980)) which allow us to quantify (even roughly) the jump expected in the sound velocity according to the thermodynamic laws. We found that the jump expected within a purely electronic BCS picture is at least three order of magnitude lower than what we observed. Note that the reduction of the electronic specific heat observed with the magnetic field by G.D. Khattack and al., Phys. Rev. B, 18, 6178 (1978) will in deed increase this difference. Clearly these uncertainties cannot explain the order of magnitude of difference that we deduced from our analysis.

Here we have attributed this difference to a change in the lattice specific heat at the α -transition a common effect for CDW transition phases. This scenario should be confirmed, for example, by a measurement of the specific heat at the α -transition which is clearly beyond the scope of the present manuscript.

We have amended the text in order give more details about the provenance of the numbers used in our analysis. The text in the main manuscript reads now as follow:

A previous experiment has quantified the magnitude of $\frac{dT_c}{dP}$ ($\frac{dT_c}{dP} \approx 0.3 \text{ K.GPa}^{-1}$) [36]. Here, we detect a jump in sound velocity as large as $\frac{\Delta v}{v_0} \approx 5.10^{-5}$. Assuming $\frac{dT_c}{dP} \approx \frac{dT_c}{dP_3}$, the expected jump of specific heat by Equation 2 is three orders of magnitude larger than the experimentally-resolved electronic specific heat ($\gamma_{el} \approx 14 \mu\text{J.K}^{-2}.\text{mol}^{-1}$ [37]. This discrepancy can be either the result of a strong anisotropy in the strain dependence of T_c or a jump in the specific heat provided by lattice. This is discussed in more detail in section F of the SM.

and in the SM as follow:

*The parameter dT_c/dP can be deduced from the study of the pressure dependence of the α -transition, as reported by Iye and al. [20]. According to this work, in the zero pressure limit, $dT_c/dP(B) = \alpha * (1 - B^*/B_c) T_c$ where $\alpha = 0.029 \pm 0.001$, $B^* = 105\text{T}$. At $T = 4.2\text{K}$ the critical field is 30T, giving a $dT_c/dP = -3e-4 \text{K.bar}^{-1} = -0.3 \text{ K.GPa}^{-1}$. Combining with our measurement of the jump of the sound velocity v , $\Delta v/v \sim 10^{-5}$, we can estimate the jump of the heat capacity ($\Delta C_p(T_c)$) at the α transition through Eq. (1). We found that $\frac{\Delta C_p(T_c)}{T_c} \approx 30 \text{ mJ.K}^{-2}.\text{mol}^{-1}$. This is three order of magnitude higher than the electronic contribution ($\Delta C_p^{ele} \approx \gamma_e T_c$) where $\gamma_e \approx 14 \mu\text{J.K}^{-2}.\text{mol}^{-1}$ is the Sommerfeld coefficient of graphite [22,23]. We note that a decrease of the Sommerfeld coefficient has been reported in presence of magnetic field which will indeed enhanced this difference [24].*

[16] The discussion of Iye et al. is all with respect to the SWMc band structure, which is does not account for the field dependence of the Landau level structure of graphite due of the neglect of strong correlation effects.

Given the highly anisotropic elastic and electronic properties of graphite, the elastic properties may respond as a witness to transitions driven by electron correlation effects (in the presence of the field dependent Landau level structure), and be renormalized by those correlations. The emphasis on the importance of the lattice may be misplaced.

Leaving aside these reservations I would urge the authors in any case to take another look at the drafting of the final two paragraphs particularly the penultimate paragraph. [minor point: The word pattern appears four times in five lines.] The first and last sentence of this paragraph are very vague.

A sharper discussion of how the experimental observations of this paper lead to the proposal of an in-plane DW would be of great interest and enhance the paper.

The referee is right, as it is often the case with semi-metals, the elastic properties are intimately related with a change in the electronic structure : a small lattice change can deeply affect the electronic spectrum (see for example the electronic and lattice structure of bismuth). As such our ultrasound measurement only proof that the lattice is sensitized to this electronic phase but does not proof that they are driving by the electron-phonon interaction.

This is why the Figure 3 of the manuscript is important to us since it underlines the fact that the instabilities first manifest in the in-plane resistance and then in the c-axis transport measurement. This fact has been known since the works Yaguchi and al. (ref [23]) but to our knowledge has never been discussed in the context of an in-plane DW state as we speculated in the last paragraph of the manuscript. Based on the referees comment we have simplified the penultimate paragraph. It reads now as follow:

Until now, the role of the electron-phonon interaction in the formation of the field induce states of graphite has not been considered. Yet it has been shown that in a Hartree-Fock approximation a DW state with a modulation vector parallel to the magnetic field can be accompanied by an in-plane modulation pinned down by the electron-phonon interaction [37,38]. We thus speculate that the α -transition corresponds to an electron-phonon assisted in-plane DW state concomitant with a lattice distortion, which would only affect R_{xx} . At the β -transition, the c-axis conductance is drastically reduced which could be due, for example, to a change in the c-axis pattern of the in-plane DW phase.

List of changes made to the main text

Title : Thermodynamic signatures of the field-induced states of graphite

p.1 column 1:

Old : Here we present the study of elastic properties of Highly Oriented Pyrolytic Graphite (HOPG) through ultrasound spectroscopy measurements.

New : Here we present the study of elastic properties of Highly Oriented Pyrolytic Graphite (HOPG) through ultrasound measurements.

p.2 column 1: In view of comments [4] for Ref. 1 and [1] of Ref.3 it appeared that we must add a few words about what the ultrasound probes, especially how can it be sensitive to both thermodynamic and dynamic aspects of the transitions. For this purpose, we worked on a new formulation of our introduction to the ultrasound properties we measured:

In the low frequency limit, or static limit, the sound velocity v_{33} measured here, corresponding to a longitudinal mode propagating along the c-axis of the hexagonal lattice, is given by $v_{33} = \sqrt{\frac{c_{33}}{\rho}}$, where ρ the mass density and c_{33} the corresponding elastic constant, using Voigt notation. In the static limit, the sound velocity is hence a second derivative of the free-energy, as specific heat or thermal expansion. As such the sound velocity is a thermodynamic probe [21]. At finite frequency, ω , dispersion effects do renormalize the sound velocity, because of coupling of the sound with internal degrees of freedom. Hence at a second order phase transition we expect two contributions to the sound velocity. First, the thermodynamic discontinuity in the elastic constant, linked to the discontinuities in the isobaric specific heat and thermal expansion through the Ehrenfest relations. The other contribution comes from the coupling to fluctuations and relaxation of the order parameter with characteristic lifetime τ [22]. The static or thermodynamic limit is achieved when $\omega\tau \ll 1$. The distinction between those two contributions will prove useful below.

p.2 column 2 :

Old: These anomalies lead to the new phase diagram presented in the semi-log plot in Fig. 2.

New: In combination with resistivity measurements (shown in SM) the ultrasound measurements reported here lead to a detailed phase diagram of the high field states of graphite presented in the semi-log plot in Fig. 2.

p.3 column 1:

Old: Below we describe the new information that our thermodynamic probe yields

New: Below we describe the new information that the ultrasound attenuation and velocity yield

p.3 column 2:

The LK theory has been successfully applied since then to various kinds of transitions ranging from liquid/gas[28], nematic/smectic [29], ferroelectric [30] and ferromagnetic transitions [31] (see [28] for a review).

Figure 2:

We added a sentence to the caption in order to improve the description of the data on the plot.

The field scales $B_{0,\alpha}$ and $B_{0,\beta}$ correspond to the ultrasound anomaly at the α and β transition extrapolated at zero frequency using the Landau-Khalatnikov formula (see text). Values reported for γ transition are taken at 255 MHz, and the δ transition shows a negligible frequency dependence.

p.4 column 1:

In figure , panel b, we see that the location of the attenuation maximum associated with the β transitions have a similar frequency dependence to that of the α transition. This indicates that the β transition, which do not show hysteresis within our resolution, is a second order transition.

p.4 column 2:

- For $T < 4.2$ K the anomalies in the sound velocity at the α and β transition loses amplitude as temperature is decreased; as shown in Fig. b. This is due to the fact that as the critical temperature for the transitions decreases, the thermodynamic contribution decreases through reduction of the specific heat.

- Nonetheless, former in-plane magnetoresistance studies [9,23] identified temperature independent anomalies close to $B \approx 45$ T (where R_{xx} changes of slope) and have been associated with a theoretically predicted lock-in transition of a CDW order along the c -axis [23]

- Our observation of a large change in the elastic properties inside the dome A supports the lock-in scenario of Ref. [23]

- Alternatively the hysteresis at the γ -transition could originate from pinning of the CDW.

p.5 column 2 :

A previous experiment has quantified the magnitude of $\frac{dT_c}{dP}$ ($\frac{dT_c}{dP} \approx 0.3$ K.GPa $^{-1}$) [36]. Here, we detect a jump in sound velocity as large as $\frac{\Delta v}{v_0} \approx 5.10^{-5}$. Assuming $\frac{dT_c}{dP} \approx \frac{dT_c}{dP_3}$, the expected jump of specific heat by Equation 2 is three orders of magnitude larger than the experimentally-resolved electronic specific heat ($\gamma_{el} \approx 14\mu\text{J.K}^{-2}.\text{mol}^{-1}$ [37]. This discrepancy can be either the result of a strong anisotropy in the strain dependence of T_c or a jump in the specific heat provided by lattice. This is discussed in more detail in section F of the SM.

Until now, the role of the electron-phonon interaction in the formation of the field induce states of graphite has not been considered. Yet it has been shown that in a Hartree-Fock approximation a DW state with a modulation vector parallel to the magnetic field can be accompanied by an in-plane modulation pinned down by the electron-phonon interaction [38,39]. We thus speculate that the α -transition corresponds to an electron-phonon assisted in-plane DW state concomitant with a lattice distortion, which would only affect R_{xx} . At the β -transition, the c -axis conductance is drastically reduced which could be due, for example, to a change in the c -axis pattern of the in-plane DW phase.

List of changes made to the Supplementary Material :

page 2 of the Supplementary Material, section C

f_0 can also be determined using Fig. 5c of the main text. At 5 K we get $f_0 \approx 3$ GHz, slightly higher than the value determined from the temperature dependence. This indicates that the fluctuation spectrum is different when the ground state of the system is controlled with a non-thermal parameter.

page 5 of the Supplementary Material :

The parameter dT_c/dP can be deduced from the study of the pressure dependence of the α -transition, as reported by Iye and al. [20]. According to this work, in the zero pressure limit, $dT_c/dP(B)=\alpha*(1-B^*/B_c)T_c$

where $\alpha=0.029 \pm 0.001$, $B^*=105\text{T}$. At $T=4.2\text{K}$ the critical field is 30T , giving a $dT_c/dP=-3\text{e-}4\text{K.bar}^{-1}=-0.3\text{K.GPa}^{-1}$. Combining with our measurement of the jump of the sound velocity v , $\Delta v/v \sim 10^{-5}$, we can estimate the jump of the heat capacity ($\Delta C_p(T_c)$) at the α transition through Eq. (1). We found that $\frac{\Delta C_p(T_c)}{T_c} \approx 30\text{ mJ.K}^{-2}.\text{mol}^{-1}$. This is three order of magnitude higher than the electronic contribution ($\Delta C_p^{ele} \approx \gamma_e T_c$) where $\gamma_e \approx 14\text{ }\mu\text{J.K}^{-2}.\text{mol}^{-1}$ is the Sommerfeld coefficient of graphite [22,23]. We note that a decrease of the Sommerfeld coefficient has been reported in presence of magnetic field which will indeed enhanced this difference [24].

Reviewers' comments:

Reviewer #1 (Remarks to the Author):

Regarding my point [2]. I agree with the authors that "below the cutoff frequency of the fluctuation spectra" the velocity feature is expected to be pure thermodynamic (static) — however Authors did not comment on their evidence that gamma-transition is below (or above) such cutoff in their measurement. Author's discussion in the paper (and their response) identifies gamma-transition as first order. By their nature, first order transitions are rarely accompanied by strong fluctuation signature — yet Authors observe the strong attenuation feature at the gamma-transition. I would like to see a more convincing evidence that gamma-transition is not second order. Furthermore, two other aspects of the observed signature of the gamma transition (that I mentioned in my previous response) — the upturn (rather than downturn) in velocity and comparable magnitude of the upturn in velocity and the increase in attenuation — have not been addressed in Author's response. These two are, again, hard to reconcile with the first order (non-fluctuating) character of the gamma-transition (as suggested by Authors)

Regarding my point [3] . it is still not clear whether Authors correctly identify the cause for the apparent 1000-fold mismatch in Eq. 2. It is not even clear whether such mismatch is real, because Authors do not have direct data for jump in heat capacity ΔC — instead they rely on its assumed proportionality to the metallic density of states. I expect that Authors can comment on how BCS-like relation between the two can be justified in this system. Authors did not address this concern in their response (or in the revised text).

Regarding the second part of my point [3] and my point [4] . I cannot follow their argumentation. Their discussion of "lattice" and "phonons" seem either highly speculative or, instead, addressed to an audience more knowledgeable about the subject than myself.

Reviewer #3 (Remarks to the Author):

The authors have considered the comments of the referees and have responded to these and made some changes.

As I wrote in my initial report, the novelty of these measurements, their timeliness, and the importance of the topic definitely merits publication in Nature Communications. The key thing is that these are measurements using ultrasound as a new probe to study the density wave instabilities induced by strong magnetic fields in graphite, in the 1D dispersing Landau levels, a problem is of fundamental importance and is as yet unsolved.

Nevertheless, I still have reservations about the lack of rigor/clarity/precision/quality in the scientific presentation which remains in certain places in the revised manuscript. In the light of that, and to fulfil my responsibilities as referee, I will simply record these reservations/comments below.

I suggest that it is particularly important that the conclusions and claims of the paper (last two paragraphs) are more clearly stated . See B below.

I wish to make clear that I do not want to contribute to a delay in publication. However I hope the authors will take the opportunity to further improve the manuscript, for the benefit of readers of the published paper, so they can better understand the results, the claims made and significance.

Obviously the title of the SM should be changed to match that of the paper.

Numbering refers to authors' response to referees.

A. Referee 1 [2], Referee 3 [8] See also detailed comments not reproduced in author response letter.

Both referees ask, with reasons, for more discussion of the γ transition and its frequency dependence. Both are calling for an improved discussion in terms of viscoelastic theory [central to the interpretation of an ultrasound study as a function of temperature and measurement frequency]. The comparison with the α , β transitions is likely to be informative to distinguish between the nature of these transitions. The γ transition is the strongest feature in the observations [Fig 1]. Yet the systematics of the data shown in Fig. 4 and Fig. 5 is not discussed, apart from a one sentence statement "The γ transition differs.....". How? And what aspects are of most interest? There may not be a complete understanding at this time. But in an experimental paper a more complete discussion of the key results is desirable.

B. Both Referee 1 and Referee 3 have issues with the discussion section of the paper.

(i) Both have reservations about the "discrepancies" claimed in the discussion (p5) of the Ehrenfest relations, because of lack of empirical knowledge of the appropriate specific heat. "The large amplitude of the jump in the sound velocity at the α transition suggests that the lattice is also involved in the instability" is the most important point, and well made. This opens the questions: what is the nature of the heat capacity anomaly at the DW instability?; what is the c-axis strain dependence of the T_c (you would expect high anisotropy, because of elastic anisotropy of graphite and because DW instability is along c-axis)? I think it would be better to simply raise these questions in such general terms in the paper, for further experimental and theoretical study. I suggest to relegate all discussion of Ehrenfest relations to the SM.

(ii) Both referees have reservations about the discussion of the role of the lattice. The paragraph "Until now..." is a shortened version of the corresponding paragraph in the original version. It is extremely unclear to me, more so than the original version. Sorry, but I don't understand the last two sentences, and am confused about what is being proposed about DWs along c-axis and in plane, at α , β transitions.

I encourage the authors, and think it is important, to make a few simple, unambiguous statements and speculations/hypotheses, so that the main claims of the paper are clear. Let me pose the following questions:

1. Do the authors accept that both the α , β transitions correspond to DW instabilities along c-axis? It is established that the q-vector evolves with field (see arXiv:1411.3323)? Or not? Either way, make the claim clear.
2. It may be the case that these transitions are accompanied by lattice distortion along c-axis? If so make that suggestion.
3. I think the authors are suggesting an accompanying in-plane density wave modulation of the c-axis DWs? If so be clear about the experimental observation that motivates that hypothesis. Be clear about what is proposed at both the α transition and β transition.

And the statement "According to our measurements the DW state forms in the vicinity of a LL depopulation, a limit where the interactions have the strongest influence", surely does not accurately reflect the phase diagram.

End of review.

Postscript note.

There is an inaccurate statement in the authors' response, on which I would like to comment. "While it was shown theoretically that electron interaction renormalized the Landau spectrum and played a role in the stabilization of the DW, it was never evidenced experimentally". Wrong. The

characteristic field of LL depopulation and lock-in transition are determined experimentally by magnetotransport, and can be only (and well) explained by including electron correlations (see arXiv:1411.3323). This also applies to 75T feature (end of second dome of Fig. 2 of this manuscript, discovered by Fauque, LeBoeuf et al [Phys. Rev. Lett. 110, 26601, 2013]), but unfortunately wrongly interpreted in that work in terms of an incorrect LL scheme.

Answers to referee 1 :

We thank referee 1 for her/his report. We report below our answers to her/his questions and the modification of our manuscript based on his comments.

Regarding my point [2]. I agree with the authors that below the cutoff frequency of the fluctuation spectra the velocity feature is expected to be pure thermodynamic (static) however Authors did not comment on their evidence that gamma-transition is below (or above) such cutoff in their measurement. Authors discussion in the paper (and their response) identifies gamma-transition as first order. By their nature, first order transitions are rarely accompanied by strong fluctuation signature yet Authors observe the strong attenuation feature at the gamma-transition. I would like to see a more convincing evidence that gamma-transition is not second order. Furthermore, two other aspects of the observed signature of the gamma transition (that I mentioned in my previous response) the upturn (rather than downturn) in velocity and comparable magnitude of the upturn in velocity and the increase in attenuation have not been addressed in Authors response. These two are, again, hard to reconcile with the first order (non-fluctuating) character of the gamma-transition (as suggested by Authors)

Response :

The referee is right. Our current data set does not reach the thermodynamic limit for the γ -transition. Therefore, we cannot reach a definite conclusion. However, we would like to comment on the following statement "By their nature, first order transitions are rarely accompanied by strong fluctuation signature yet Authors observe the strong attenuation feature at the gamma-transition." Consider the sequence of CDW instabilities in 2H-TaSe₂. There, a second order phase transition towards an incommensurate CDW at 121K is followed by a first order transition from an incommensurate (IC) to commensurate (C) CDW phase at 96K. A dramatic increase of the ultrasound velocity [M. H. Jerico PRB 22 4907 1980] and of the internal friction [M. Barmatz PRB 12 4367 1975] occurs at the IC to C transition, and are attributed to the lock-in of the CDW phase on the lattice. Despite the fact that the origin of the large attenuation is unclear and perhaps related to discommensuration domains of the CDW [M. Barmatz PRB 12 4367 1975], the empirical signatures of a first order lock-in transition are very similar to what is observed in graphite at the γ transition. Indeed in graphite the largest ultrasound response (both sound velocity and attenuation) is observed at the γ transition and it shows hysteresis. Moreover the interpretation of the γ transition as a lock-in transition agrees with the scenario proposed by F. Arnold and al and is supported by referee 2. Therefore we find it reasonable to suggest that this transition could be a first-order lock-in transition. We note that the word used by referee 1 is *rarely* and not *never*.

Based on the referees comment we removed the term "first order" for the γ -transition in the manuscript except at the end of the paragraph on the γ -transition where it appears as a suggestion (supported by all the arguments made above) *Furthermore, by comparing $\frac{\Delta v(B)}{v_0}$ measured in field-up and field-down sweeps (shown in Fig.5 d)), we find that the γ -transition is characterized by an hysteresis loop, which possibly suggests that it is of a first order nature, contrary to the α transition. We note however that we did not reach the static limit for this transition. The hysteresis could alternatively originate from other effects such as the pinning of a CDW.*

Regarding my point [3] . it is still not clear whether Authors correctly identify the cause for the apparent 1000-fold mismatch in Eq. 2. It is not even clear whether such mismatch is real, because Authors do not have direct data for jump in heat capacity ΔC instead they rely on its assumed proportionality to the metallic density of states. I expect that Authors can comment on how BCS-like relation between the two can be justified in this system. Authors did not address this concern in their response (or in the revised text).

We do agree with the referee that our estimation is only putative since we currently don't know the amplitude of the jump in the specific heat at the transition. Yet it sounds to us reasonable to discuss the amplitude of the jump in the sound velocity using a BCS-like expression for the jump in C/T . There are, at least, two facts which suggest that this transition can be described within a BCS picture:

(i) the link between the critical field (B_0) and temperature (T_0) where $T_0 = T^* \exp(-B^*/B_0)$ (T^* is a temperature scale associated with the Fermi energy, and B^* a field associated with the Landau level

dispersion along the field direction) corresponds to a BCS-type formula for mean-field-type pairing transitions as established by Yoshioka et Fukuyama ([8]). This link is experimentally verified on almost two orders of magnitude (from 10 to 0.3K).

(ii) The amplitude of the gap (Δ) as deduced from the c-axis resistance in Kish graphite at 47T and 64T (see ref [22] on page 3) follows a BCS-like law : $\frac{2\Delta}{k_B T_c} = 3.4$. Also we wish to underline that in the case of the superconducting transition of $\text{SrTi}_{1-x}\text{Nb}_x\text{O}_3$ (a low carrier superconductor $n=2e20\text{cm}^{-3}$) the BCS relation between the jump in the specific heat and the gamma-term holds even if the carrier concentration is low (X. Lin, PRB 90, 140508(R) (2014)). Within this picture the jump in the specific at T_c is related to the normal state specific heat. The proportionality factor can range from 1.4 to about 3 in the strong coupling limit but it would not explain the fact that the observed sound velocity anomaly is 1000 times bigger than what expected within this purely electronic BCS picture.

Regarding the second part of my point [3] and my point [4] . I cannot follow their argumentation. Their discussion of lattice and phonons seem either highly speculative or, instead, addressed to an audience more knowledgeable about the subject than myself.

Since both referees expressed reservations regarding our discussion of the Ehrenfest relation we have transferred it to the SM section. Moreover, in the current version, remarks on the amplitude of the sound velocity are all grouped in a single paragraph at the end of the discussion section. We emphasize the experimental fact that in comparison to other known cases of Density Wave instabilities, the detected jumps in the sound velocity are large. In our opinion, this is the main evidence in favor of a role played by the lattice in this field-induced phase transition. The text now reads as:

The amplitude of the sound velocity jump at the α -transition is an order of magnitude smaller than the one found at the SDW transition of PF_6 [33] and two orders of magnitude smaller than the one found at the incommensurate CDW state of 2H-TaSe_2 [32] but it is surprisingly high for a system with a carrier concentration four orders of magnitude smaller than these metals. As discussed in section E of the SM, the large amplitude of the sound velocity anomaly measured at the α -transition suggests that the lattice degrees of freedom are also involved in the DW instability. This unexpected contribution rises several questions and call for further works. What is the amplitude of the specific heat jump at the DW transitions and the respective contribution of electrons and phonons ? Our measurement of c_{33} indicates that the DW instabilities are coupled to c-axis strain. Could the electron-phonon interaction also induce an in-plane lattice deformation along with the c-axis nesting process as suggested by early theoretical works [37,38]? Could an in plane lattice deformation explain why the α -transition appears in R_{xx} but only weakly in R_{zz} as shown in Fig.3 ?

Finally, let us underline that invoking the role played by electron-phonon interaction in the formation of the DW phase does not express a particularly uncommon opinion. CDW transitions are generally believed to be accompanied by lattice distortions (see for example M. D. Johannes and I. I. Mazin, Phys. Rev. B 77, 165135 (2008)). The point of our discussion is to link our observation of a large elastic response to a probable (or at least plausible) role of the lattice.

Answers to referee 2 :

We thank referee 3 for her/his report. We report below our answers to her/his questions and the modification of our manuscript based on his comments.

A. Referee 1 [2], Referee 3 [8] See also detailed comments not reproduced in author response letter. Both referees ask, with reasons, for more discussion of the γ -transition and its frequency dependence. Both are calling for an improved discussion in terms of viscoelastic theory [central to the interpretation of an ultrasound study as a function of temperature and measurement frequency]. The comparison with the α , β transitions is likely to be informative to distinguish between the nature of these transitions. The γ -transition is the strongest feature in the observations [Fig 1]. Yet the systematics of the data shown in Fig. 4 and Fig. 5 is not discussed, apart from a one sentence statement "The γ transition differs.... "How? And what aspects are of most interest? There may not be a complete understanding at this time. But in an experimental paper a more complete discussion of the key results is desirable.

We thank the referee for his comment on the γ -transition. Following his advice we have changed the part on the γ -transition. The text reads as follow : *The γ -transition differs from other transitions regarding its temperature and frequency dependence. In contrast wto the α and β -transition, the γ -transition is almost temperature independent much like the δ -transition (see Fig. 4 a) and b)). But its frequency dependence (reported on Fig. 5) b)) differs from the δ -transition (which is frequency independent). As the field increases we observe a hardening at the γ -transition i.e, an opposite behaviour compared to the α -transition. The γ -transition differs as well in its signature in transport properties. As seen in Fig.1, the γ -transition has, by far, the largest effect on the elastic properties of graphite, while its signature in R_{zz} is small compared to the others transitions.*

B. Both Referee 1 and Referee 3 have issues with the discussion section of the paper. (i) Both have reservations about the discrepancies claimed in the discussion (p5) of the Ehrenfest relations, because of lack of empirical knowledge of the appropriate specific heat. The large amplitude of the jump in the sound velocity at the ? transition suggests that the lattice is also involved in the instability is the most important point, and well made. This opens the questions: what is the nature of the heat capacity anomaly at the DW instability?; what is the c-axis strain dependence of the Tc (you would expect high anisotropy, because of elastic anisotropy of graphite and because DW instability is along c-axis)? I think it would be better to simply raise these questions in such general terms in the paper, for further experimental and theoretical study. I suggest to relegate all discussion of Ehrenfest relations to the SM.

We have followed the comment of the referee 3 and we replied to it in our last answer to the referee 1.

(ii) Both referees have reservations about the discussion of the role of the lattice. The paragraph Until now is a shortened version of the corresponding paragraph in the original version. It is extremely unclear to me, more so than the original version. Sorry, but I dont understand the last two sentences, and am confused about what is being proposed about DWs along c-axis and in plane, at α , β -transitions.

I encourage the authors, and think it is important, to make a few simple, unambiguous statements and speculations/hypotheses, so that the main claims of the paper are clear. Let me pose the following questions:

1. Do the authors accept that both the α , β transitions correspond to DW instabilities along c-axis? It is established that the q-vector evolves with field (see arXiv:1411.3323)? Or not? Either way, make the claim clear. 2. It may be the case that these transitions are accompanied by lattice distortion along c-axis? If so make that suggestion. 3. I think the authors are suggesting an accompanying in-plane density wave modulation of the c-axis DWs? If so be clear about the experimental observation that motivates that hypothesis. Be clear about what is proposed at both the ? transition and ? transition.

We thank referee 3 for his/her comments. All the questions raised are important and interesting. Here are our answers and the modification of our manuscript based on those useful comments.

(1) We do agree with referee 3 that the scenario invoking a nesting instability along c-axis (as expected

by field-induced one-dimensionality and proposed by Fukuyama decades ago), remains the most probable scenario. Indeed, as pointed out in arXiv:1411.3323, this picture implies a field-dependent nesting vector. In our manuscript, we had stated that the γ -transition is consistent with a lock-in scenario implying a field dependent q-vector (at least in the case of one of the two transitions). To emphasize this point, we added the following sentence in the γ -transition section: *Our observation of a large change in the elastic properties inside dome A supports the lock-in scenario of Ref. [9] implying a field dependence of the wave vector modulation of the DW along the c-axis .*

(2) Our measurements of c_{33} indeed indicate that a lattice distortion along c-axis occurs along with the DW transitions.

(3) We cannot exclude that the DWs also occurs along with an *in-plane* lattice distortion. However to make such an assertion we would need additional measurements. Nonetheless the resistivity measurement motivates this hypothesis. As illustrated in Figure 3 of our manuscript, at low temperature, the first anomaly occurs in R_{xx} and at slightly higher field in R_{zz} . This fact may suggest that the α -transition is accompanied by an in-plane distortion that might cause a bigger response in R_{xx} than in R_{zz} . Referee 3 invites us to make an unambiguous statement about the exact sequence of transitions and lattice distortions. However, adding a lattice degree of freedom significantly increases the number of possible scenarios. In our opinion, the discussion about the possible scenarios is beyond the scope of the paper and will require further experiments such as diffraction studies to identify the wave vector of the DW phase.

The paragraph starting by "Until now..." was originally written as speculative thoughts on the role that the electron-phonon interaction could play in setting the orientation of the DW wave-vector, which has never been considered so far in the literature. Since that paragraph was not essential to the manuscript, and given the reservations expressed by referee 1, we have explicitly reformulated the ideas as opened questions that will motivate further theoretical and experimental work: *Our measurement of c_{33} indicates that the DW instabilities are coupled to c-axis strain. Could the electron-phonon interaction also induce an in-plane lattice deformation along with the c-axis nesting process as suggested by early theoretical works [37,38]? Could an in plane lattice deformation explain why the α -transition appears in R_{xx} but only weakly in R_{zz} as shown in Fig.3 ?*

And the statement According to our measurements the DW state forms in the vicinity of a LL depopulation, a limit where the interactions have the strongest influence, surely does not accurately reflect the phase diagram.

To address the referees concern, we have amended the sentence. In the new version, it reads: *According to our measurements, the highest critical temperature occurs in the vicinity of a LL depopulation. This is where the density of states is largest and therefore the effect of interactions is expected to be stronger [J. Alicea and al., Phys. Rev. B 79, 241101(R) (2009)].*

Postscript note. There is an inaccurate statement in the authors response, on which I would like to comment. While it was shown theoretically that electron interaction renormalized the Landau spectrum and played a role in the stabilization of the DW, it was never evidenced experimentally. Wrong. The characteristic field of LL depopulation and lock-in transition are determined experimentally by magnetotransport, and can be only (and well) explained by including electron correlations (see arXiv:1411.3323). This also applies to 75T feature (end of second dome of Fig. 2 of this manuscript, discovered by Fauque, LeBoeuf et al [Phys. Rev. Lett. 110, 26601, 2013]), but unfortunately wrongly interpreted in that work in terms of an incorrect LL scheme.

We respectfully disagree with this postscript note. In this manuscript we report the first measurement of a LL depopulation above the critical temperature. All the measurements and the analysis in arXiv:1411.3323 has been done in the field-induced state where the LL depopulation is hidden by the collapse of the dome A. As such their result cannot be taken as a direct evidence of a LL depopulation in contrast to our measurement.

REVIEWERS' COMMENTS:

Reviewer #1 (Remarks to the Author):

Regarding the gamma-transition discussion. I understand the Author's position on the matter, but I think the current discussion of the gamma-transition does not explore all the information that this measurement provide. I want to point out again to a piece of evidence about gamma-transition in Figure 1: the attenuation (at the peak) and the magnitude of the velocity upturn are of comparable magnitude when both expressed in frequency units. This naturally points to the possibility that these two quantities are related via Kramers-Kronig relation — as velocity shift and attenuation are near continuous (second order) phase transition.

I want to comment on Eq. 1 and accompanying discussion. Although as a phenomenological argument it parallels directly the discussion of the temperature-tuned second-order phase transitions due to Landau and Khalatnikov, an important distinction is that magnetic field — unlike temperature is a vector. It is a good guess that B in eq. 1 is not absolute value: e.g., it has been shown in [Scientific Reports 7, 1733 (2017), ref. [12] in the manuscript] that B_0 (for gamma-transition) is strongly dependent on magnetic field orientation. It therefore seems that in the modified form of eq. 1 both B_0 and f_0 will be dependent on field orientation. This in particular means that relaxation times discussed later in the manuscript can vary significantly with field orientation. Although this manuscript is not concerned with angular dependences, it may still be worthwhile to extend the discussion of Eq. 1 in this way.

comment on the "On the other hand, the transition shows up as a change of slope in the sound velocity. This indicates that the transition involves a different order parameter - strain coupling." at the top of Page 4. All transition, including those associated with the continuous change of the order parameter may (and typically do) have an abrupt change in slope. The discontinuity in velocity across beta transition can simply be too small to be observed in this measurement.

In the second column of page 5:

"As discussed in section E of the SM, the large amplitude of the sound velocity anomaly measured at the alpha-transition suggests that the lattice degrees of freedom are also involved in the DW instability."

"While the wave vector direction of the DW state remains to be determined, our thermodynamic data and analysis indicate that both the electron-electron and the electron-lattice interactions should be accounted for in theoretical models."

It is not at all clear what these sentences attempt to convey. The pulsed echo measurements are essentially dc measurements. In particular, they do not provide access to the relevant phonon wavelengths and frequencies where effects of electron-phonon and electron-electron physics might be distinguished experimentally. Hence, it would help if the text of the manuscript be very clear that "lattice" and "phonons" discussion refers to theoretical modeling of possible mechanisms of symmetry changes rather than direct consequences of the data presented in this work.

Reviewer #3 (Remarks to the Author):

The authors have responded constructively to the suggestions of both referees 1 and 3, whose concerns were similar in a number of places.

The expanded discussion on the γ transition is welcome.

Moving material to the supplementary material adds focus and clarity to the discussion. The changed text achieves an improved balance between the presentation and discussion of new results, and a clearer formulation of open, yet unresolved issues. These improvements will better help the community to both understand the point of view of the authors of this manuscript, challenge it as they see fit, and stimulate further experimental and theoretical work.

The back and forth of this reviewing process, which has been extensive and generally constructive, should not be further protracted in my opinion.

Confirmation of aspects of the phase diagram (Fig 2), by the ultrasound technique is important. The confirmation of the lock-in [γ] transition reported in ref 9 is valuable. The speculation of in-plane density modulation should stimulate further work. These results will also contribute to the resolution of the nature of the DW instabilities in this system.

I recommend publication of the manuscript.

I would like to add some thoughts. The authors' response to the postscript note of my last review is just not right. The collapse of the dome (disappearance of DW instability) and the depopulation of the LL (hosting that DW instability) are the same thing. Ref 9 resolves for the first time the emptying of both the (0, up) and (-1, down) LLs [not resolved in the authors' present ultrasound study].

Three of the authors (LeBoeuf, Fauque and Behnia) were co-authors of Phys. Rev. Lett. 110, 26601 (2013), which discovered the B-dome (a beautiful result) but unfortunately got the scheme of occupied Landau levels (each of which potentially hosts a DW instability until it becomes depopulated with increasing field) completely wrong. This, and the conversation of the refereeing process, leads me to emphasize that the new experimental data, and extensive theoretical modelling of ref 9 has clarified this issue. That theory also calculates the field-dependent DW gaps for graphite; the cited reference to Alicea and Balents discusses bismuth.

None of this other prior work diminishes the importance of the results reported in this manuscript. In my opinion, and as stated in my original review, the novelty, timeliness of the data and importance of the topic definitely merits publication in Nature Communications.

But since our common aim is scientific progress, this should be achieved by putting all the pieces of the jigsaw together, respecting a diversity of opinion. Therefore a close study of the detail of ref 9 is something I respectfully recommend to the authors.

In spite of all that, I recommend publication of the manuscript in its current form, modulo details of spelling etc. which I won't comment on here.

Answers to referee 1 :

We thank referee 1 for her/his report. We report below our answers to her/his questions and the modification of our manuscript based on his comments.

Regarding the gamma-transition discussion. I understand the Authors position on the matter, but I think the current discussion of the gamma-transition does not explore all the information that this measurement provide. I want to point out again to a piece of evidence about gamma-transition in Figure 1 : the attenuation (at the peak) and the magnitude of the velocity upturn are of comparable magnitude when both expressed in frequency units. This naturally points to the possibility that these two quantities are related via Kramers-Kronig relation as velocity shift and attenuation are near continuous (second order) phase transition.

Response :

Following the comment of referee 1 we let open the conclusion on the nature of γ transition in the manuscript. We end the subsection γ by the following sentence : *Note however that we did not reach the static limit for this transition. Alternatively, the γ transition could be of second order character with the hysteresis loop originating from other effects such as the pinning of the CDW.*

I want to comment on Eq. 1 and accompanying discussion. Although as a phenomenological argument it parallels directly the discussion of the temperature-tuned second-order phase transitions due to Landau and Khalatnikov, an important distinction is that magnetic field unlike temperature is a vector. It is a good guess that B in eq. 1 is not absolute value: e.g., it has been shown in [Scientific Reports 7, 1733 (2017), ref. [12] in the manuscript] that B_0 (for γ -transition) is strongly dependent on magnetic field orientation. It therefore seems that in the modified form of eq. 1 both B_0 and f_0 will be dependent on field orientation. This in particular means that relaxation times discussed later in the manuscript can vary significantly with field orientation. Although this manuscript is not concerned with angular dependences, it may still be worthwhile to extend the discussion of Eq. 1 in this way.

We do agree with referee 1 that it would be of high interest to study the angular dependence of f_0 and compare it with that of B_0 . Following his comment, we added the following sentence: *Note that in this experiment the field is oriented along the c-axis. It would be interesting to study the angular dependence of f_0 and compare it with that of B_0 [12]."*

comment on the On the other hand, the transition shows up as a change of slope in the sound velocity. This indicates that the transition involves a different order parameter - strain coupling, at the top of Page 4. All transition, including those associated with the continuous change of the order parameter may (and typically do) have an abrupt change in slope. The discontinuity in velocity across beta transition can simply be too small to be observed in this measurement.

We have modified our statement on the β -transition and acknowledge the comments of referee 1. We have added the following sentence : *On the other hand, the (β)-transition shows up as a change of slope in the sound velocity. The absence of a discontinuity can be either due to the limitation in the experimental resolution or an indication that this transition involves a different order parameter- strain coupling.*

In the second column of page 5:

As discussed in section E of the SM, the large amplitude of the sound velocity anomaly measured at the alpha-transition suggests that the lattice degrees of freedom are also involved in the DW instability.

While the wave vector direction of the DW state remains to be determined, our thermodynamic data and analysis indicate that both the electron-electron and the electron-lattice interactions should be accounted for in theoretical models.

It is not at all clear what these sentences attempt to convey. The pulsed echo measurements are essentially dc measurements. In particular, they do not provide access to the relevant phonon wavelengths and frequencies where effects of electron-phonon and electron-electron physics might be distinguished experimentally. Hence,

it would help if the text of the manuscript be very clear that lattice and phonons discussion refers to theoretical modeling of possible mechanisms of symmetry changes rather than direct consequences of the data presented in this work.

We agree with referee 1 that our measurement does not probe directly the phonon spectrum. According to our analysis at the α -transition, the jump in the specific heat is much larger than the (zero field) electronic specific heat, which suggests that this transition is accompanied by a lattice deformation.

As discussed in section E of the SM, the large amplitude of the sound velocity anomaly measured at the α -transition suggests that the lattice degrees of freedom are also involved in the DW instability. by As discussed in section E of the SI, the large amplitude of the sound velocity anomaly measured at the α -transition suggests that, in the formalism of the Ginzburg-Landau theory, the order parameter is coupled to a lattice strain along the c-axis. This unexpected coupling raises several questions and calls for further works.

As we write at the end of our discussion, the lattice deformation can either occur along the c-axis or in the plane perpendicular to the magnetic field, as theoretically discussed in papers [37,38] where the electron-phonon interaction pin the in-plane modulation vector. We thus conclude that our result invite to consider both the electron-electron and electron-phonon interactions to understand the nature of the field induced state in graphite. To clarify our statement we have added the following sentence in the last paragraph : *While the wave vector direction of the DW state remains to be determined, our thermodynamic analysis indicate that the order parameter is coupled to a c-axis strain, suggesting that the interaction between quasiparticles and the lattice is at play and should be accounted for in theoretical models.*